# Carbon dioxide electroreduction on single-atom nickel decorated carbon membranes with industry compatible current densities

Hengpan Yang[1], Qing Lin[1], Chao Zhang[1], Xinyao Yu[1], Zhong Cheng[1], Guodong Li[1], Qi Hu[1], Xiangzhong Ren[1], Qianling Zhang[1], Jianhong Liu[1] & Chuanxin He [1]*

Carbon dioxide electroreduction provides a useful source of carbon monoxide, but comparatively few catalysts could be sustained at current densities of industry level. Herein, we construct a high-yield, flexible and self-supported single-atom nickel-decorated porous carbon membrane catalyst. This membrane possesses interconnected nanofibers and hierarchical pores, affording abundant effective nickel single atoms that participate in carbon dioxide reduction. Moreover, the excellent mechanical strength and well-distributed nickel atoms of this membrane combines gas-diffusion and catalyst layers into one architecture. This integrated membrane could be directly used as a gas diffusion electrode to establish an extremely stable three-phase interface for high-performance carbon dioxide electroreduction, producing carbon monoxide with a 308.4 mA cm$^{-2}$ partial current density and 88% Faradaic efficiency for up to 120 h. We hope this work will provide guidance for the design and application of carbon dioxide electro-catalysts at the potential industrial scale.

[1] Shenzhen Key Laboratory for Functional Polymer, College of Chemistry and Environmental Engineering, Shenzhen University, Shenzhen, Guangdong 518060, China. *email: hecx@szu.edu.cn

The excessive emission of greenhouse gas $CO_2$ results in a chain of serious climate problems. As a consequence, it is intensely urgent to convert $CO_2$ resources into beneficial fuels or chemicals, and simultaneously alleviate atmospheric $CO_2$ level[1,2]. Especially, electrochemical reduction of $CO_2$ has tremendous potential to contribute value-added compounds directly from atmospheric $CO_2$, coupling with renewable-energy electricity. This research area has evolved remarkably in the past few decades, making significant progress towards industrial application[3,4].

$CO_2$ is highly stable molecule, which needs proper catalysts to activate. Therefore, catalyst development has played a crucial part with a steady stream of novel electro-catalysts, i.e., metals, metal oxides, metal complexes, transition-metal phosphides or carbon-based materials[5–12]. Some catalysts could even exhibit nearly 100% selectivity for specific products (i.e., single-atom Co sites)[13]. On the other hand, $CO_2$ electroreduction takes place around the three-phase interface, hence the diffusion and adsorption of $CO_2$ molecules, desorption of reduced products on the surface of electro-catalyst determine the reaction rate to a large extent[14]. However, a majority of catalytic materials for $CO_2$ electroreduction are still developed, characterized and evaluated in typical H-type electrochemical cells. In these H-type cells, the catalyst layer is fully immersed and $CO_2$ gas is generally bubbled into the conductive electrolyte[15]. Accordingly, the current densities of $CO_2$ conversion would be largely limited by the low solubility and slow diffusion of $CO_2$ in aqueous systems, and hardly satisfy the industrial-relevant current densities (>100 mA cm$^{-2}$) in $CO_2$ reduction[4]. To solve these problems, electrochemical reactors (like flow cell) using gas-diffusion electrode have been seriously investigated in recent years. Instead of aqueous electrolyte, $CO_2$ is supplied from the gaseous phase through a gas-diffusion layer to construct a gas–liquid interface adjoined to the solid catalyst layer. This unique architecture has the potential to deliver enough $CO_2$ directly onto the catalyst surface and reach very high current densities. To produce a usable gas-diffusion electrode, catalysts for $CO_2$ electroreduction were usually adhered or deposited on a specific substrate[16]. Unfortunately, because of the weak contacts between substrate and electro-catalyst, the electroactive species become detached and easily fall off, thus reducing the long-term performance of $CO_2$ electroreduction[17].

Herein, we report an effective method for the preparation of a high-yield and self-supported Ni single-atom/porous carbonfiber membrane catalyst (NiSA/PCFM) via electrospinning method for high-efficiency $CO_2$ electroreduction. The extensive single-atom Ni sites distributing throughout carbon nanofiber could play a decisive role in activation of $CO_2$. The porous and interconnected carbon nanofibers of NiSA/PCFM provide substantial channels for $CO_2$ diffusion and electron transport. In addition, NiSA/PCFM has favorable mechanical strength and flexibility, which can be easily tailored to specific shapes or thickness. Therefore, NiSA/PCFM could combine gas-diffusion and catalyst layers into a single architecture. This integrated NiSA/PCFM membrane could be immediately utilized as GDE compartment for high-performance $CO_2$ reduction, avoiding the weak connection between the substrate and electro-catalyst.

## Results

### Synthesis and characterization of NiSA/PCFM catalysts.
The fabrication method of NiSA/PCFM membrane is described in Fig. 1a, which involves electrospinning process and heated treatment. The primary fibers were firstly electrospun from the mixture solution of ZIF-8 nanoparticles (Supplementary Fig. 1), Ni(NO$_3$)$_2$·6H$_2$O and polyacrylonitrile (PAN). Notably, ZIF-8 nanoparticles did not contain Ni ions and mainly acted as pore former in our experiments. Ni ions were introduced by Ni

(NO$_3$)$_2$·6H$_2$O and would be reduced by carbonized organic polymers of PAN in the following heated treatment, leaving atomically dispersed Ni atoms anchored on nitrogen-doped porous carbon. It also deserves to be mentioned that the NiSA/PCFM membranes yielded in our laboratory apparatus are at least ~280 cm$^{-2}$ in size (Fig. 1b). The manufacture method is uncomplicated and easily adapted to larger size (or thickness) membranes. The NiSA/PCFM membrane is robust and flexible shown in Fig. 1b, which could retain the initial structure during bending stress (Supplementary Fig. 2). As disclosed by the stress–strain curve in Fig. 1c, NiSA/PCFM membrane also can endure ~1.3 MPa tensile strength. Besides, the water contact angle (CA) of NiSA/PCFM catalyst was very close 0°, indicating the hydrophilic property (Supplementary Fig. 3). If a little amount of Nafion mixture was spread on the surface of NiSA/PCFM, water CA was measured to be 142.7° ± 3.0° (Fig. 2a), implying the excellent hydrophobic property. This hydrophobic property could guarantee that NiSA/PCFM is not easily flooded in aqueous solution, which is crucial for GDE devices[18].

NiSA/PCFM was also characterized with morphological and gas adsorption experiments. The field-emission scanning electron microscopy (FE-SEM, Fig. 2b, c, Supplementary Fig. 4a, b) and high-resolution transmission electron microscopy (HRTEM, Fig. 2d) showed that the nanofibers of NiSA/PCFM created three-dimensionally interconnected architecture with visible and well-distributed macropores (ca. 100 nm) throughout these nanofibers. N$_2$ sorption isotherms (Fig. 3a) furtherly verifies that NiSA/PCFM has a 714 m$^2$ g$^{-1}$ specific surface area and hierarchical structure. The pore size distribution peaks at ca. 100 nm and ca. 1 nm (~3.6 times of the dynamic diameter of $CO_2$ molecules)[19] reveal both macropores and micropores. The macropore channels could enhance the gas transportation to micropores where $CO_2$ molecules are mainly captured, causing the remarkable $CO_2$ adsorption capacity of NiSA/PCFM under atmospheric pressure (Fig. 3b). This exceptional structure could accelerate the diffusion and adsorption of $CO_2$, resulting in $CO_2$ enrichment around the active sites. If no ZIF-8 nanoparticles but only Ni(NO$_3$)$_2$ were added into the precursor solution for electrospinning, as-synthesized NiSA/CFM presents typical microporous structure and no macropores were observed on the nanofibers (Supplementary Fig. 5, 6a–b), which sharply reduced the $CO_2$ adsorption capacity (Fig. 3b) even with a higher specific surface area (789 m$^2$ g$^{-1}$, Supplementary Table 1) than that of NiSA/PCFM. PCFM has the similar hierarchical structure (Supplementary Figs. 6c, d, 7) and $CO_2$ adsorption capacity as NiSA/PCFM.

In addition, based on the orientation of the distorted graphene layers in HR-TEM images (Supplementary Fig. 4d), a longer range orientation of the graphene unit is seen in NiSA/PCFM. This might affect the conductivity and redox reactions, which involve electron transfer through the electrode surface[20]. Moreover, the Ni, N, and C elements were homogenously distributed and no obvious accumulation of Ni was founded, revealed via energy-dispersive X-ray spectroscopy elemental mapping (EDS, Fig. 2e). Aberration-corrected high-angle annular dark-field scanning transmission electron microscopy (HAADF-STEM) can directly distinguish Ni element from those adjacent light elements[21]. As presented in Fig. 2f and Supplementary Fig. 8, the white spots distributed on the black background of NiSA/PCFM and NiSA/CFM are proposed to be isolated Ni atoms, due to the difference of Z contrast of Ni, N, and C elements[22].

The elemental status of NiSA/PCFM were further explored by multiple methods. Only two peaks centered at 26.2° and 44.0° are recognized in the XRD patterns, which can be indexed to the (002) and (100) planes of carbon[13]. The diffraction peaks belonging to metallic Ni or Ni oxides are absent in all three

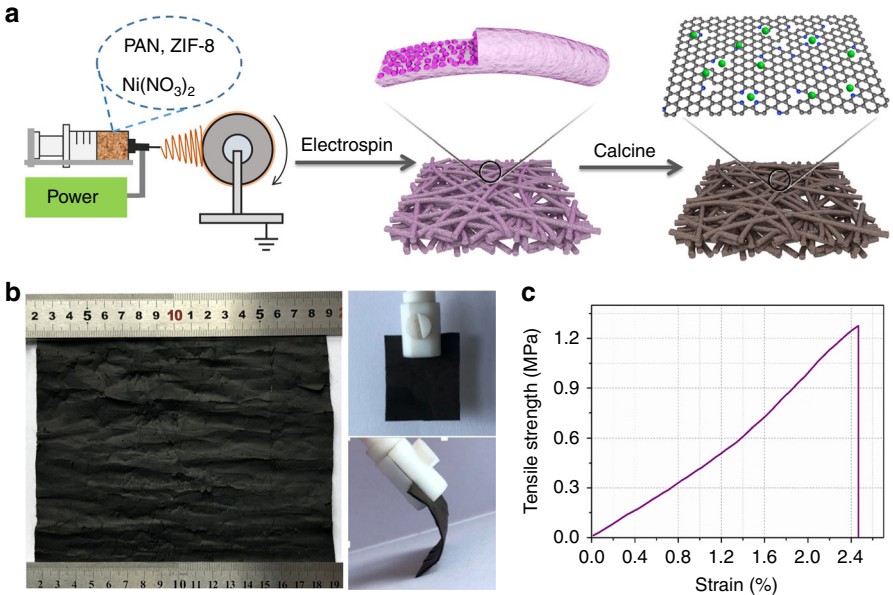

**Fig. 1 Physical characterizations of NiSA/PCFM. a** Synthesis strategy of NiSA/PCFM. **b** Digital images and **c** stress-strain curve of NiSA/PCFM membrane.

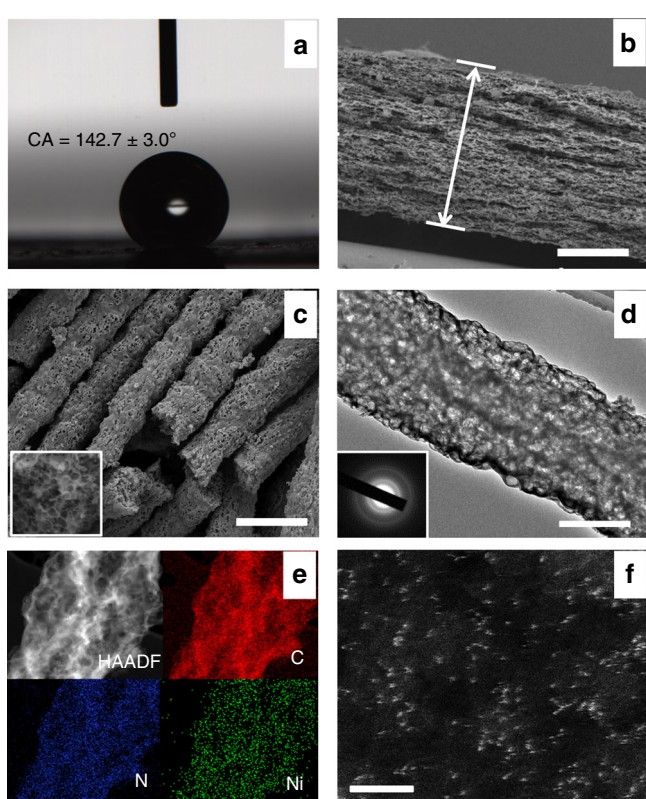

**Fig. 2 Morphological and structural characterizations of NiSA/PCFM.**
**a** The water contact angles of NiSA/PCFM spraying with a little amount of Nafion solution. **b** Cross-sectional and **c** high-resolution SEM images of NiSA/PCFM. **d** HR-TEM image of NiSA/PCFM, the inset of (**d**) displays the lattice fringe. **e** EDS mapping of an independent NiSA/PCFM nanofiber. **f** Magnified HAADF-STEM images of NiSA/PCFM, those white dots are supposed to be Ni single atoms. Scale bars, 100 μm (**b**), 1 μm (**c**), 0.5 μm (**d**), and 2 nm (**f**).

patterns (Supplementary Fig. 9). The relatively sharp and intensive peaks at 26.2° and 44.0° for NiSA/PCFM, compared with that of NiSA/CFM, indicate that the additional ZIF-8 nanoparticles result in a higher degree of crystallization and graphitization. Furthermore, X-ray photoelectron spectroscopy (XPS, Supplementary Table 2) N *1s* spectrum of NiSA/PCFM (Fig. 3c) could deconvolute into pyridine N (398.2 eV), Ni–N (399.4 eV), pyrrole (400.5 eV), graphitic (401.3 eV) and oxidized (403.0 eV) N species. The binding energies of Ni $2p_{3/2}$ in NiSA/PCFM (Supplementary Fig. 10) were reflected by the peak at 854.8 eV, locating between metallic $Ni^0$ (853.5 eV) and $Ni^{2+}$ (855.8 eV), suggesting that the Ni atoms in NiSA/PCFM is likely to be in a low-valent state[21]. The coordination environment of Ni species would be intensively investigated via X-ray absorption fine spectroscopy (XAFS). Figure 3d exhibits the X-ray absorption near-edge structure (XANES) spectrum in the Ni K-edge of NiSA/PCFM, using Ni foil and NiO as the references. The near-edge spectra of NiSA/PCFM locates between Ni foil and NiO, implying that the valence state of those isolated Ni atoms are between metallic ($Ni^0$) and oxidized ($Ni^{2+}$) status[22,23]. The Fourier transform (FT) k3-weighted χ(k) function of the extended X-ray absorption fine structure (EXAFS) spectra of NiSA/PCFM exhibited a dominant peak at 1.42 Å for Ni–N coordination (Fig. 3e). As contrast, Ni foil shows a typical Ni-Ni pair at 2.20 Å, and the Ni–O interaction of NiO sits around 1.62 Å. According to the fitting results, the proposed local structure of NiSA/PCFM involves coordination by four N atoms ($Ni–N_4$, Fig. 3f, Supplementary Table 3)[24–27]. NiSA/CFM also possess $Ni–N_4$ structure, indicated by XPS (Supplementary Fig. 11) as well as XAFS spectra (Supplementary Fig. 12). No Ni element was observed in XPS patterns of PCFM (Supplementary Fig. 13). The Ni amount of NiSA/PCFM and NiSA/CFM were 1.3 wt% and 1.2 wt%, revealed by inductively coupled plasma optical emission spectroscopy.

**$CO_2$ electro-reduction with H-type cell**. On the basis of above characterizations, as-prepared NiSA/PCFM composites possess excellent physical properties and massive single-atom doping, which may facilitate enhanced activity for $CO_2$ electroreduction. The $CO_2$ electro-reduction activity, NiSA/PCFM, NiSA/CFM, and PCFM powders were firstly drop-casted onto a carbon paper using typical powdering and binding method for LSV tests and electrolysis in H-type cell. These powdered samples were named as P-NiSA/PCFM, P-NiSA/CFM and P-PCFM, easy to distinguish. The anode and cathode sections were separated by a

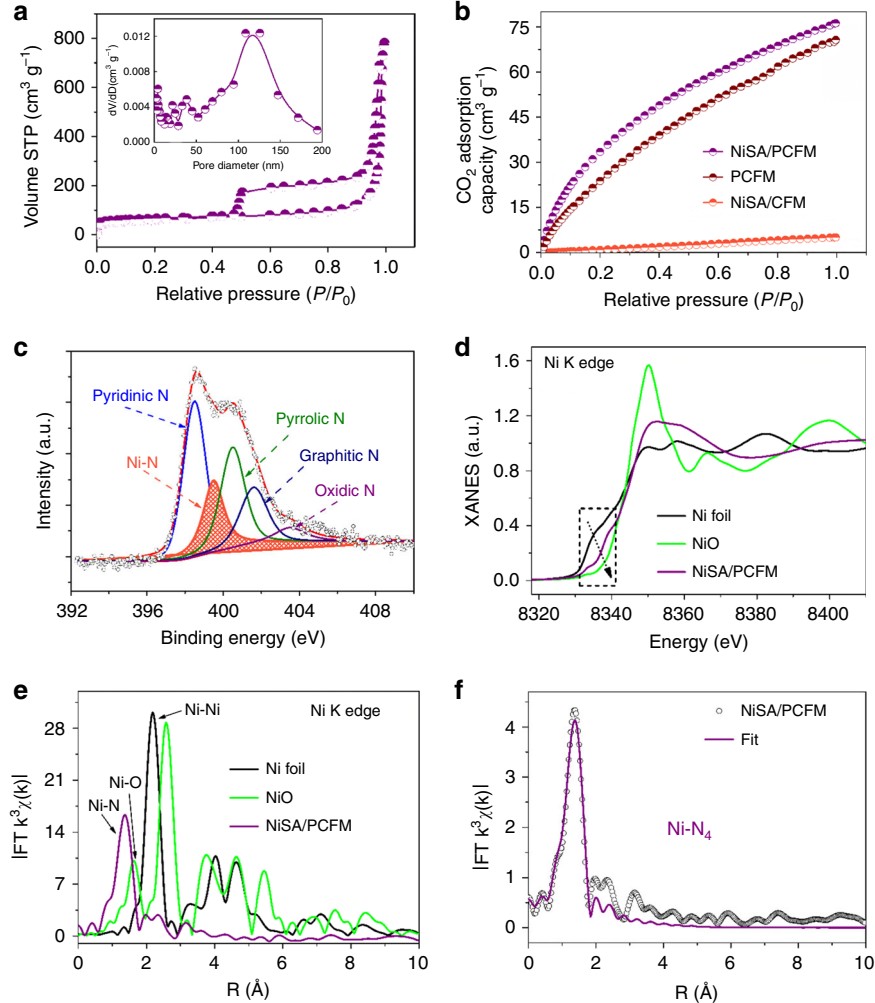

**Fig. 3 Chemical analysis of NiSA/PCFM. a** $N_2$ sorption isotherms of NiSA/PCFM, inset shows the pore size distribution. **b** $CO_2$ adsorption amount of various catalysts. **c** N 1s XPS spectra of NiSA/PCFM. **d** XANES spectra at the Ni K-edge of Ni foil, NiO and NiSA/PCFM; **e** The Fourier transform of EXAFS data for three samples; **f** Fitting for the EXAFS data of NiSA/PCFM, inset is the Ni-$N_4$-C structure, Ni (green sphere), N (blue sphere), and C (gray sphere).

Nafion-117 proton exchange membrane. The cathodic compartment was continuously bubbled with $CO_2$ and injected directly into gas chromatography (GC) for quantitative detection of gaseous products. Liquid-phase products were determined by $^1H$ NMR spectroscopy. The primary products catalyzed by three catalysts are only CO and $H_2$, and no trace of any liquid products were observed in $^1H$ NMR spectroscopy (Supplementary Fig. 14). Clearly, the P-NiSA/PCFM and P-NiSA/CFM maintain excellent faradic efficiency (FE) for CO generation at working potentials from −0.5 to −1.0 $V_{RHE}$, and P-NiSA/PCFM could achieve a maximum value of 95% at −0.7 $V_{RHE}$ (Fig. 4a, Supplementary Fig. 15a, b). In contrast, P-PCFM catalyst without Ni composition showed very limited FEs for CO production throughout the potential range (Supplementary Fig. 15c), indicating that the Ni-$N_4$ structure played a dominating role in $CO_2$ electroreduction to CO. As mentioned above, NiSA/PCFM membrane has excellent mechanical strength and could directly acted as working electrode in H-type cell system. As displayed in Fig. 4c, the best CO Faradaic efficiency obtained by NiSA/PCFM is 96% at −0.7 $V_{RHE}$, similar to that of P-NiSA/PCFM (95%). However, NiSA/PCFM brought forth a 56.1 mA $cm^{-2}$ CO partial current density at −1.0 $V_{RHE}$ (Fig. 4d), significantly higher than those of P-NiSA/PCFM, P-NiSA/CFM and P-PCFM (Fig. 4b).

To understand the observed activity difference of three catalysts, density functional theory (DFT) calculations were performed on the well-defined N–C and Ni–$N_4$–C structures (Supplementary Fig. 16, Table 4). The corresponding free energy profiles (ΔG) are summarized in Fig. 5a. According to the simulations, the conversion of $CO_2$ to adsorbed COOH* intermediate on N–C and Ni–$N_4$–C structures are uphill in the free energy profiles, demonstrating that the first electronic step is the rate-determining step for both systems[28]. Especially, the Ni–$N_4$–C has more favorable ΔG (0.7 eV) than N–C (1.3 eV) for this step. The lower ΔG requires less energy to form *COOH intermediate, consistent with the much smaller Tafel slope of NiSA/PCFM (117 mV/dec) and NiSA/CFM (156 mV/dec), than that of PCFM (204 mV/dec, Fig. 5b)[29,30]. Notably, the partial CO current density catalyzed by P-NiSA/PCFM and NiSA/PCFM are higher than those of PCFM and NiSA/CFM (Fig. 4b). NiSA/ PCFM also presented a much larger overall current density in the linear sweep voltammetry (LSV) tests (Supplementary Fig. 17). Since NiSA/PCFM and NiSA/CFM possess very close amount of Ni single atoms, the differences in $CO_2$ electroreduction activity may also be ascribed to the hierarchically porous structures[10,31,32]. Different from the typical microporous structure of NiSA/CFM, NiSA/PCFM shows hierarchically porous structures, which can accelerate the diffusion and adsorption of reactants, lead to remarkable $CO_2$ adsorption capacity (Fig. 3b) and a higher electrochemical active surface areas (ECSA, Supplementary Fig. 18). In a word, the free-standing and

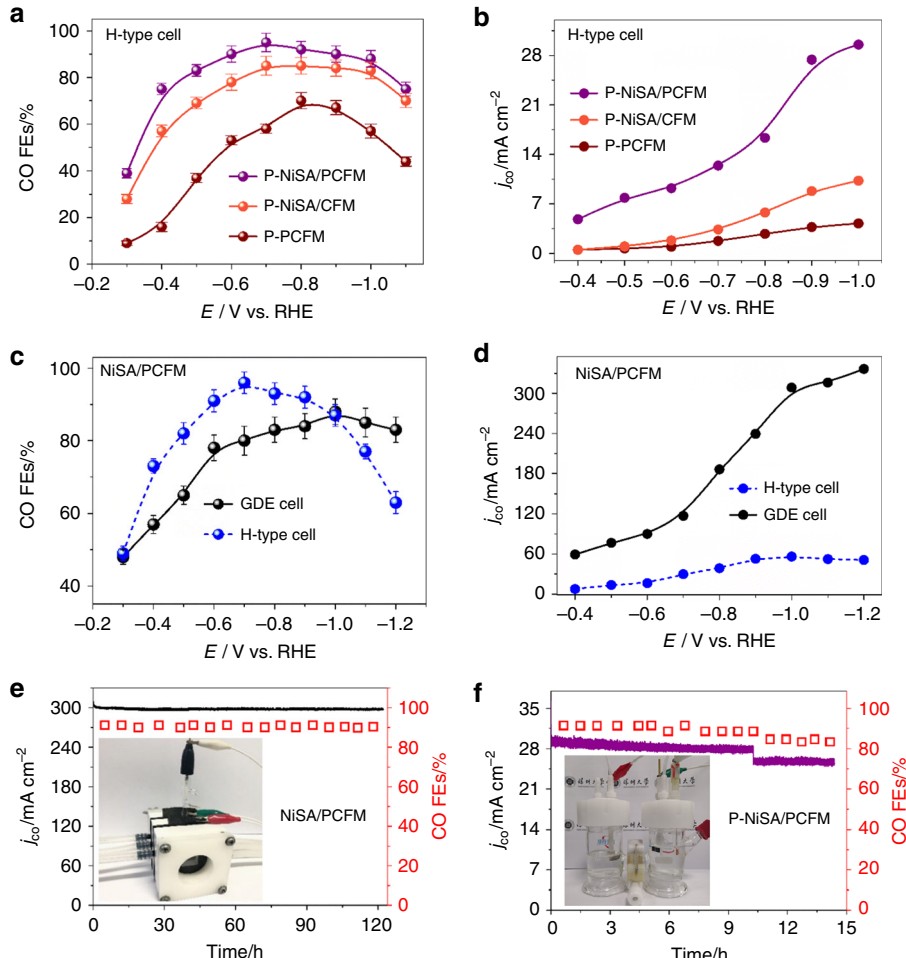

**Fig. 4 Electrocatalytic CO$_2$ reduction. a** CO faradaic efficiency and **b** partial current densities for three catalysts at various cathode potentials in H-type cell. **c** CO faradaic efficiencies, **d** partial current densities of NiSA/PCFM at various cathode potentials in different cells. **e** Long-term stability tests in GDE cell and **f** H-type cell at –1.0 V$_{RHE}$, respectively.

hierarchically porous carbon structure of NiSA/PCFM membrane could generate plenty of effective Ni single atoms, which actually participate in CO$_2$ reduction. Another important advantage of the NiSA/PCFM relates to the lower electric resistance displayed in electrochemical impedance spectroscopy (EIS, Supplementary Fig. 19), in favor of fast charge transport in CO$_2$ electrolysis[5].

**CO$_2$ electro-reduction with GDE cell.** All the above electrolysis were conducted in a conventional H-type cell, thereby the catalyst layer is completely soaked in a conductive CO$_2$-saturated electrolyte. The current densities were severely restricted by mass transportation as a result of the limited solubility of CO$_2$ in aqueous solution[16]. Therefore, we deployed a flow cell equipped with GDE device (see Supplementary Fig. 20 and Fig. 5c) and tested the NiSA/PCFM catalyst to advance toward industrial current densities (>100 mA cm$^{-2}$). Firstly, NiSA/PCFM, NiSA/CFM, and PCFM were loaded onto a gas diffusion layer (SIGRACET) via typical methods[33] as the cathode compartment (denoted as P-NiSA/PCFM) in flow cell for LSV tests. As presented in Supplementary Fig. 17d, e, the current densities of three samples observed in flow cell are all far greater than those in H-type cell, indicating the superiority flow cell device.

Figure 4c, d displays the CO FEs and partial current densities plotted against the RHE-corrected cathode potential. H$_2$ and CO accounts for up to 99% of the transferred charge on NiSA/PCFM

membrane using GDE cell, while no liquid product could be detected (Supplementary Fig. 15d), similar to those in H-type cell. In GDE device, It is also apparent that the NiSA/PCFM can be performed exceeding its optimized range of cathode potentials in the above H-type cell measurements. The CO FEs of NiSA/PCFM begin to reduce obviously at more negative potentials than −0.7 V in H-type cell, but more than 80% FEs could still be obtained even at −1.2 V$_{RHE}$ in GDE cell (Fig. 4c). Besides, partial CO current densities with GDE device were significantly higher. For example, with the same 88% CO FEs, NiSA/PCFM exhibits 308.4 mA cm$^{-2}$ CO partial current density in GDE cell at −1.0 V$_{RHE}$, which is more than 5 times larger than that of H-type cell (56.1 mA cm$^{-2}$). In addition, the partial CO current density obtained in GDE cell could reach an even higher value of 336.5 mA cm$^{-2}$ with 83% CO FE at −1.2 V$_{RHE}$. To the best of our knowledge, accounting for the electrochemical conversion of CO$_2$ to CO, the catalytic activity of NiSA/PCFM is comparable to these state-of-the-art electro-catalysts in previously reports (Supplementary Table 5), especially at industrial-relevant current densities.

To explore the stability of the NiSA/PCFM membrane, long-term tests were conducted using NiSA/PCFM catalysts at −1.0 V$_{RHE}$ electrode potential where the best CO FE was acquired. CO FEs and partial current densities against time are plotted in Fig. 4e. Both CO FEs and partial current densities of the NiSA/PCFM membrane showed only a negligible drop during 120 h tests, retaining more than 95% of initial value. More than

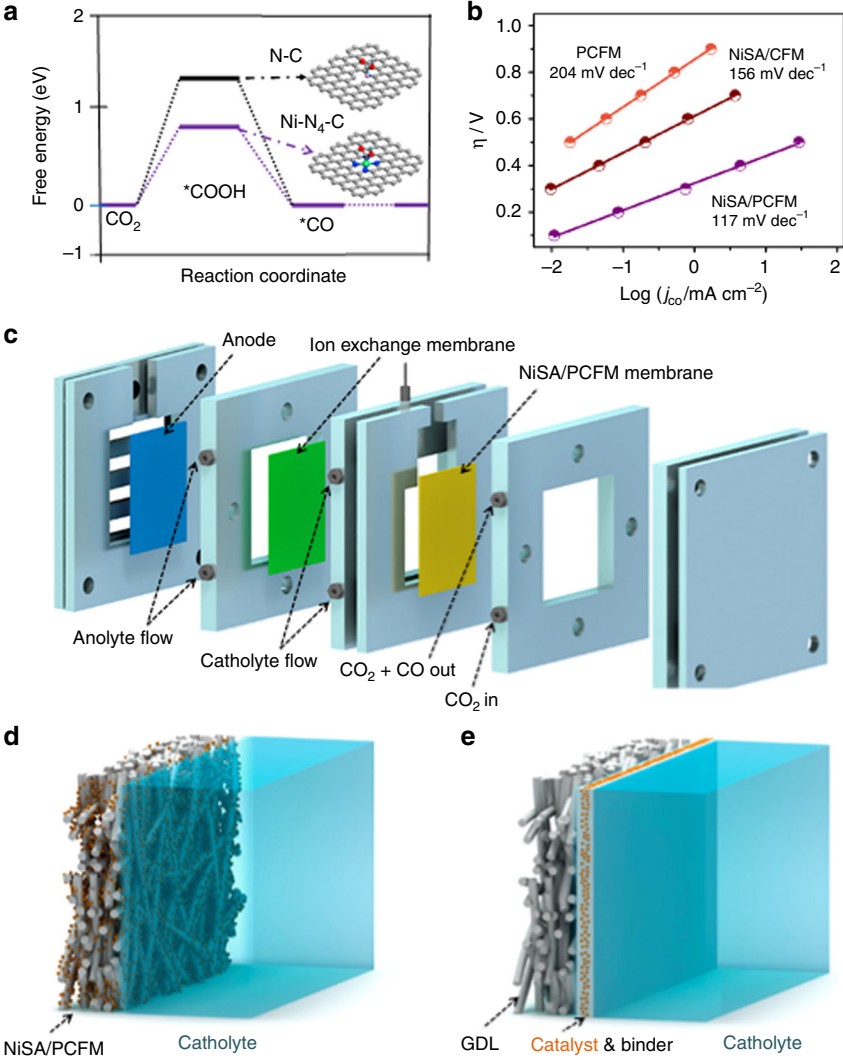

**Fig. 5 Proposed mechanism of NiSA/PCFM. a** Free energy diagram of $CO_2$ to adsorbed *COOH intermediate on N-C and $Ni-N_4-C$ doped graphene structure; **b** Comparisons of Tafel plots. **c** Graphic illustration of a GDE device; Schematic for **d** NiSA/PCFM membrane directly used as GDE and **e** typical GDE cell with catalyst powder loaded onto a gas-diffusion layer via polymer binder.

100 h stability of NiSA/PCFM membrane could also be achieved in two-electrode system (Supplementary Fig. 21)[34,35]. Moreover, the CO FEs obtained with P-NiSA/PCFM was very close to those of NiSA/PCFM using cathode potentials from $-0.3\ V_{RHE}$ to $-1.2\ V_{RHE}$, attaining an optimal value of 87% at $-1.0\ V_{RHE}$ (Supplementary Fig. 22a). Furthermore, P-NiSA/PCFM could preserve a lower but steady current density of 70.5 mA cm$^{-2}$ at $-0.6\ V_{RHE}$ cathode potential for 2 h. However, current densities dropped significantly at more negative $-0.8\ V_{RHE}$ or $-1.0\ V_{RHE}$ cathode potential within the short time of 1 h (Supplementary Fig. 22b, d), which is noticeably worse than the long-term stability of NiSA/PCFM. In a typical GDE device, the catalysts (e.g., P-NiSA/PCFM) were usually spray-coated (or deposited) onto a gas diffusion layer using polymer binders (GDL, Fig. 5c). During electrolysis procedure, massive gas products would pour out from the active sites along with the large current densities at relatively negative cathode potentials. Consequently, the electro-active species would easily fall off and reduce the long-term performance of $CO_2$ reduction, due to the weak contacts between catalyst and substrate[16]. If P-NiSA/PCFM was tested in H-type cell, the current-time curve also shows a distinct drop after 10 h electrolysis (Fig. 4f). As mention early, NiSA/PCFM has abundant Ni single atoms and cross-linked carbon nanofibers, which

combines both gas-diffusion and catalyst layers. Hence, NiSA/PCFM membrane was readily used as cathode compartment (Fig. 5d, e), instead of spray-coating the catalyst ink onto a specific gas-diffusion layer. The integrated structure and fine mechanical strength of NiSA/PCFM membrane could guarantee the long-time stability under high current densities.

Besides, the hydrophobicity of gas-diffusion and catalyst layer also have vital impact on the long-term stability[18]. The thickness of catalyst layer is always in nanoscale and this thin catalyst layer could easily be flooded by electrolyte and destroy the three-phase interface around the active sites[4]. In our system, the thickness of NiSA/PCFM membrane is in the range of hundreds of micrometers (Fig. 2b), which is not easily to be totally soaked by electrolyte and establish an extremely stable three-phase interface for high-performance $CO_2$ electroreduction. Furthermore, the architecture of NiSA/PCFM membrane still remained unchanged after electrolysis (Supplementary Fig. 23, 24) and could be readily utilized for next run with simple oven dry.

## Discussion

In conclusion, we prepared an effective approach for the relatively large-scale production of single-atom Ni doped porous carbon

membrane. We started from the screening tests in typical H-type cell, which could produce CO with maximum 96% faradaic efficiency at $-0.7$ $V_{RHE}$. DFT calculations based on *COOH intermediate provide a molecular understanding of the observed high efficiency on single-atom Ni sites. In addition, the outstanding structure of NiSA/PCFM could generate substantial effective Ni single atoms actually participating in $CO_2$ reduction. Moving to GDE device, this NiSA/PCFM membrane can combine gas-diffusion as well as catalyst layer to create a reactive and provide stable three-phase interface for high-performance $CO_2$ electroreduction. Remarkably, NiSA/PCFM membrane could manifest CO product with a commercially relevant partial current density of 308.4 mA cm$^{-2}$ as well as a high Faradaic efficiency of 88% at $-1.0$ $V_{RHE}$ for at least 120 h, which is even comparable to those state-of-the-art electro-catalysts for $CO_2$ reduction to CO. This work may also give some reference and guidance for the design and preparation of integrated electro-catalysts utilized in gas-diffusion device.

## Methods

**Chemicals and Characterizations**. All reagents were used as received.

LSV tests were performed using an electrochemical workstation (Princeton Applied Research 263 A) with a three-electrode system in typical divided electrochemical cell separated by Nafion®117 (Dupond) membrane between cathode and anode. Gas phase products were analyzed by Gas Chromatography (SRI 8610 C). Microstructure images were taken using field emission scanning electron microscope (FEI JEOL-7800F). High-resolution TEM images as well as element mapping were obtained using transmission electron microscope (JEM-2100F). Aberration-corrected HAADF-STEM characterization was carried on a JEOL JEM-ARF200F TEM/STEM. X-ray diffraction spectra were taken by a Rigaku MiniFlex 600 powder diffractometer. $N_2$ adsorption-desorption was performed using a Micromeritics ASAP 2460 instrument. $CO_2$ adsorption was carried out using Quantachrome Autosorb-IQ2-MP. X-ray photoelectron spectra were taken with ThermoVG Scientific ESCALAB 250 X-ray photoelectron spectrometer. The X-ray absorption find structure spectra, including XANES and EXAFS at Ni K-edge, were collected at 1W1B station in Beijing Synchrotron Radiation Facility.

**Synthesis of ZIF-8**. ZIF-8 crystals were prepared via rapidly pouring an aqueous solution (16 mL deionized water) of $Zn(NO_3)_2 \cdot 6H_2O$ (7.9 mmol, 2.34 g) into an aqueous solution (160 mL deionized water) of 2-methylimidazole (553 mmol, 45.4 g). Then, the mixed solution magnetically stirred for 5 min at room temperature. After stirring, the mixture was filtered by filter paper. The liquid product was collected by centrifuge (7000 revolutions per minute, 30 min) and thoroughly cleaned by deionized water for at least five times. The product was dried at 60 ºC for 12 h in a vacuum drying oven.

**Synthesis of NiSA/PCFM, NiSA/CFM, and PCFM**. NiSA/PCFM were synthesized using an electrospinning process. First, 1.5 g of polyacrylonitrile (PAN, Mw $=130,000$), 1.5 g of ZIF-8 and 30 mg of $Ni(NO_3)_2 \cdot 6H_2O$ powder were dissolved in 15 mL of N,N-dimethylformamide (DMF) via vigorously stirring to get a homogenous mixture. The transparent mixture was diverted into a plastic syringe with a stainless needle at the tip. After electrospinning procedure, the as-spun raw fibers were firstly preoxidated in air at 250 °C for 1 h and immediately carbonized under argon gas at an optimum temperature of 900 °C for 2 h. The heating rate was 5 °C min$^{-1}$. After heat treatment, the residue was cooled to room temperature under argon gas flow. Then, the resultant materials were immersed into $H_2SO_4$ solution (3.0 M) for 10 h to remove the remaining Zn species, Ni or NiO nanoparticles. NiSA/PCFM membrane was thus obtained.

The transparent solution containing 30 mg $Ni(NO_3)_2 \cdot 6H_2O$ and no ZIF-8 was also electrospun and carbonized under 900 °C to act as control for the comparison. The resulting product was referred to as NiSA/CFM.

**$CO_2$ reduction procedure. H-type cell**. Electrolysis and LSVs was performed in a in conventional H-type electrochemical cell separated by Nafion®117 membrane between cathode and anode, consisted of Pt foil as the counter electrode and an Ag/AgCl as reference electrode in $CO_2$ or $N_2$-saturated 0.5 M $KHCO_3$ solution. A specific volume of the catalyst (PCFM, NiSA/CFM or NiSA/PCFM) ink was then drop-casted on carbon paper electrode to achieve an approximately 1 mg cm$^{-2}$ loading amount and then dried at room temperature. This carbon paper would be utilized as the cathode for $CO_2$ electro-reduction.

**GDE device**. Measurements at high current densities were performed in a self-made micro flow cell (see Fig. 4e and Supplementary Fig. 20). Different from tests using H-type cell, NiSA/PCFM membranes were cut into the desired size and shape and directly served as cathode compartment, since these membranes are flexible and self-supporting. Certain amount of Nafion solution (5 wt%, Dupond) was spray-coated on this membrane to get a hydrophobic property. A commercial Pt/C electrode was used as anode and an Ag/AgCl were acted as the reference. 0.5 M $KHCO_3$ aqueous solution was utilized as electrolyte, which were separated by a piece of Nafion®117 membrane. Electrolytes were cycled at 50 mL min$^{-1}$. The $CO_2$ gas was supplied at rate of 20 mL min$^{-1}$ to the cathode and was flown through the NiSA/PCFM membrane.

## Data availability

The data that support the findings of this study are available within the paper and its Supplementary Information file or are available from the corresponding authors upon request.

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

## Acknowledgements

We sincerely appreciate the financial support from Shenzhen Government's Plan of Science and Technology (JCYJ20170817095041212 and JCYJ20170818091657056), the Natural Science Foundation of Guangdong Province (2017A040405066), National Natural Science Foundation of China (51902209, 21975162). We also acknowledge the help of Electron Microscopy Centre of Shenzhen University for testing the aberration-corrected HADDF-STEM.

## Author contributions

C.H. conceived the project and idea. H.Y. designed the experiments and wrote the manuscript. Q.L., C.Z., X.Y., Z.C., G.L. and Q.H. carried out the synthesis, material characterizations and electrochemical measurements. X.R., Q.Z. and J.L. analysed the data from these experiments and commented on the manuscript.

## Competing interests

The authors declare no competing interests.
