## [Peer Review File · Nature Communications]

Reviewers' comments:

Reviewer #1 (Remarks to the Author):

In this manuscript, the authors report the preparation of a high-yield, flexible and self-supported single-atom Ni decorated porous carbon membrane, NiSA/PCFM. This integrated NiSA/PCFM membrane could be directly used as gas diffusion electrode for CO₂ electroreduction, generating CO product with a -308.4 mA cm⁻² partial current density and 88% faradaic efficiency for up to 40 hours. This ultrahigh current density along with remarkable faradaic efficiency might stratify the potential industrial application. This manuscript also proposes an effective method for large-scale production of single-atom Ni catalyst. This paper is very interesting and written in a highly comprehensive manner. I recommend its publication after addressing the following comments.

1) Single-atom Ni catalysts were usually derived from Ni ions doped metal-organic frameworks (e.g., *J. Am. Chem. Soc.* 2017, 139, 8078–8081). The Ni ions were confined, stabilized by N species and then reduced to Ni single atoms by the surrounding carbon within the MOFs. In this manuscript, author only gave a very rough explanation that "The fabrication method of NiSA/PCFM membrane is described in Scheme 1a, which involves electrospinning process and heated treatment". More information should be presented.

2) In the synthesis procedure, plenty of ZIF-8 nanoparticles (1.5 g) and only small amount of Ni(NO₃)₂·6H₂O (30 mg) were added into the precursor solution. What role do massive ZIF-8 nanoparticles play in the preparation of NiSA/PCFM membrane, pore former or other function?

3) Zn residues might still exist in NiSA/PCFM membrane, since ZIF-8 nanoparticles contain abundant Zn species. Moreover, Zn-based materials also show catalytic activity for CO₂ electroreduction (e.g., *Angew. Chem. Int. Ed.* 2016, 55, 9297–9300). Hence, contrast experiments should be performed.

4) Magnified HAADF-STEM images of NiSA/PCFM in Figure 1f displays the well-distributed Ni single atoms. Similar HAADF-STEM images of NiSA/CFM would be also welcomed for better understanding of the Ni species.

5) Based on my understanding, Ni or NiO nanoparticles could also form under high-temperature carbonization because of the easy accumulation of Ni ions. Do Ni or NiO nanoparticles exist in NiSA/PCFM membrane?

6) NiSA/PCFM shows lower electric resistance in EIS test. Why?

7) The description of the flow cell is very vague, and an enhanced discussion would be necessary, e.g., the flow rate of both electrolyte and CO₂ gas.

8) NiSA/PCFM powder was drop-casted onto a carbon paper using Nafion binders for H-type cell, and NiSA/PCFM membrane was directly used as a GDE device for CO₂ electroreduction. This comparison is not appropriate. Contrast experiment with NiSA/PCFM powder being drop-casted onto a GDE device would be necessary.

9) The manuscript characterized and claimed excellent stability of NiSA/PCFM membrane after long-term testing, e.g., SEM, XPS, and TEM. Would NiSA/PCFM membrane fall apart after electrolysis?

10) Stability is one of the most important technique targets in CO₂ electroreduction. Both CO FEs and partial current densities of the NiSA/PCFM membrane showed only a negligible drop during 40 hours tests. It is an impressive but not fully-convincing result. Thus, stability test with longer time is welcomed.

Reviewer #2 (Remarks to the Author):

CO₂ electroreduction could provide an alternative source of CO. Unfortunately, very limited electrocatalysts could be stably operated at industrial-relevant current densities, due to the diffusion control of CO₂ in aqueous solution. This manuscript constructs a flexible single-atom Ni decorated porous

carbon membrane, which combines gas-diffusion (carbon nanofibers) and catalyst layers (Ni single atoms) into one architecture. This integrated NiSA/PCFM membrane could be directly utilized as a GDE device, producing CO with a $-308.4 \text{ mA cm}^{-2}$ partial current density and 88% faradaic efficiency for up to 40 hours. This is a nice work in the synthesis of single-atom catalysts as well as CO₂ electroreduction. Therefore, I would like to recommend the publication of this manuscript after considering the following minor comments:

(1) Several experimental techniques have been reported in recent papers for the synthesis of single-atom Ni catalysts (like Chem 2018, 4, 285; Nat. Commun. 2019, 10, 2359). Author should provide the detailed formation mechanism of this NiSA/PCFM membrane.

(2) Author mentioned that "NiSA/PCFM membranes yielded in our laboratory apparatus are at least $\sim 280 \text{ cm}^{-2}$ in size". Author should also offer the thickness and weight of this NiSA/PCFM membrane, which are equally important for an electro-catalyst.

(3) In a typical GDE device, electro-catalysts were mostly immobilized onto a conductive and porous substrate via polymer binders or electrodeposition, easily causing the fragility of the whole system. The integrated NiSA/PCFM membrane could be directly utilized as a GDE device, producing CO with more than -300 mA cm^{-2} partial current density. Author should also load NiSA/PCFM powder onto a gas-diffusion substrate, and compare the performance in CO₂ electroreduction.

(4) NiSA/PCFM membrane could produce CO with a $-308.4 \text{ mA cm}^{-2}$ partial current density and 88% faradaic efficiency for up to 40 hours. This is indeed an excellent result. It is notable that other researchers have reported the constant ethylene selectivity for an initial 150 operating hours (Science 2018, 360, 783). Hence, stability test of NiSA/PCFM membrane for more than 100 hours should be conducted.

(5) NiSA/PCFM membrane could be stably operated at industrial-relevant current densities. Would this catalysis system still be stable under large flow rate of CO₂ gas? Please clarify.

(6) Flow cell reactor was introduced in this manuscript, which is also mentioned in recent works about CO₂ reduction (Science 2019, 365, 367; Nat. Commun. 2019, 10, 2807). Intuitive description of the flow cell reactor used in this manuscript should be provided.

(7) The pH value of electrolyte has significant impact on CO₂ electroreduction. Therefore, CO₂ electroreduction should also be tested in different electrolyte (like KCl, or KOH).

Reviewer #3 (Remarks to the Author):

The manuscript, entitled "CO₂ electroreduction to CO on single-atom Ni decorated porous carbon membranes with industrial-relevant current densities", reported that a self-supported single-atom Ni decorated porous carbon membrane NiSA/PCFM was fabricated by electrospinning technology. The authors claimed that this self-supported electrocatalyst provided a stronger contact between substrate and electro-catalyst. As a result, it served as GDE directly and shows a high faradic efficiency and current density in gas-diffusion cell system. To be honest, this manuscript does not show enough contribution in both of scientific concept and technology breakthrough. Besides, the manuscript should be further revised due to the lack of enough evidence and data for some conclusions. Therefore, the reviewer thinks this work has not been met the threshold for publishing in Nature Communication.

The detail reasons are as follows:

1. The fabrication novelty of Ni single atom is similar to previous reports. As we know, ZIF derived Ni single atom electrocatalyst for CO₂RR application has recently been published over three times [J. Am. Chem. Soc. 2017, 139, 8078-8081; J. Am. Chem. Soc. 2017, 139, 14889-14892; Energy Environ. Sci., 2018, 11, 1204-1210]. Besides, the self-supported single-atom Ni decorated carbon matrix or metal free carbon membrane were also published and shown an excellent CO₂RR performance [Joule, 2019, 3, 584-594, Angew. Chem. Int. Ed., 2017, 56, 7847-7852.]. Therefore, both of scientific concept and

fabrication strategy are still undermined by the high similarity with previous reports.

2. The authors highlight the high current density of their electrocatalyst. I agree with the author's viewpoint that industrial-relevant current densities ($>100 \text{ mA cm}^{-2}$) is a significant point to judge the performance of CO₂RR electrocatalyst. However, this technologic problem has been solved by gas-diffusion cell system which has reported in many previous papers (at least 20 papers), such as, Science, 2018, 360, 783–787, Energy Environ. Sci., 2019, 12, 1442-1453, Adv. Mater. 2018, 30, 1803111, Energy Environ. Sci., 2019, 12, 1442-1453 et al.]. Therefore, this work has not contributed enough novelty in technological area of CO₂RR if it is just used in gas-diffusion cell system to achieve a high current density.

3. It is noteworthy that the highest j_{co} of NiSA/PCFM is 29.5 mA cm⁻² in H-type cell, which is quite similar to previous reports [J. Am. Chem. Soc. 2017, 139, 14889-14892, Adv. Mater. 2018, 30, 1706287]. Moreover, the comparison samples in table S4 were tested in H-type cell system but the j_{co} of NiSA/PCFM in table S4 was obtained in gas-diffusion cell system. Therefore, it is unreasonable to claim that the NiSA/PCFM shows a higher CO₂RR performance because these values came from two different test system and mass loading.

4. The authors should provide evidences to demonstrate the advantages of their NiSA/PCFM in stability and current density by additional control group. For instance, loading ZIF-derived NiSA porous carbon in commercial gas diffusion layer.

5. Generally, the calculation of CO₂RR faradic efficiency shows a $\pm 10\%$ errors, especially in gas-diffusion cell system. Therefore, it is necessary to add error bars for all faradic efficiency to ensure the accuracy of data.

6. The LSV curves of NiSA/PCFM (NiSA/CFM, PCFM) tested in the gas-diffusion cell system should be provided (not just j_{co}).

7. As shown in supporting information, the authors use $Q=I \times t$ to calculate faradic efficiency. However, different with the sealing structure of H-type cell, for the gas diffusion cell, the gas products are fluxed to GC directly because the gas is flowing and products concentration does not relevant to the quantity of electricity. Therefore, please explain the measurement and calculate method of the faradic efficiency. The calculate formula of faradic efficiency in gas diffusion cell show as follows:

8. The authors claim that the active center of NiSA/PCFM is Ni-N₄ and they demonstrate it by EXAFS and HAADF-STEM image. But the HAADF-STEM image is not clear enough to identify Ni single atom. Please give a new image to ensure the Ni atom could be observed and measured. Does the scale of Ni atom in Figure 1f match its atomic diameter well?

9. The fabrication method of PCFM should be given. I have not found this part in both of manuscript and supporting information.

10. Many previous reports show that the heteroatom doped carbon materials (such as, nitrogen) also have outstanding CO₂RR performance in both of faradic efficiency and current density [Adv. Funct. Mater. 2018, 1800499, Adv. Energy Mater. 2017, 1701456, et al]. Therefore, the percentage of N with different configuration should be provided (based on XPS). Meanwhile, a control group should be given to exclude the main contribution of N if NiSA/PCFM (NiSA/CFM, PCFM) shows an obvious difference in N contents.

11. In the manuscript, the authors use 3 M H₂SO₄ solution to remove the Zn species. However, similar with Ni-N₄, the Zn-N_x may exist in the sample due to the high ratio of Zn in ZIF even though Zn is easy to volatilize in high temperature. Therefore, the amount of Zn should be measured by ICP-AES to exclude its contribution. The authors should provide additional DFT calculation with the Zn-N_x sites in CO₂RR if Zn content is too high to ignore.

12. It should be confirmed that the high CO₂RR performance of NiSA/PCFM comes from Ni single atom, as authors' declare, through additional metal poisoning experiments.

Response to Reviewer's Comments

Response to the Reviewer 1

Point 1: *Single-atom Ni catalysts were usually derived from Ni ions doped metal-organic frameworks (e.g., J. Am. Chem. Soc. 2017, 139, 8078–8081). The Ni ions were confined, stabilized by N species and then reduced to Ni single atoms by the surrounding carbon within the MOFs. In this manuscript, author only gave a very rough explanation that “The fabrication method of NiSA/PCFM membrane is described in Scheme 1a, which involves electrospinning process and heated treatment”. More information should be presented.*

Reply and modification: Thanks for the comment. Just like you pointed out, Ni ions were usually confined (or doped) within MOFs and then reduced to Ni single atoms by the surrounding carbon ligands of MOFs (e.g., J. Am. Chem. Soc. 2017, 139, 8078). In this manuscript, the primary fibers were firstly electrospun from the mixture solution of ZIF-8 nanoparticles, $\text{Ni}(\text{NO}_3)_2 \cdot 6\text{H}_2\text{O}$ and polyacrylonitrile (PAN). Notably, ZIF-8 nanoparticles did not contain Ni ions and only acted as pore former in our experiments. Ni ions were introduced by $\text{Ni}(\text{NO}_3)_2 \cdot 6\text{H}_2\text{O}$ and would be reduced by carbonized organic polymers of PAN in the following heated treatment, leaving atomically dispersed Ni atoms anchored on nitrogen-doped porous carbon.

Point 2: *In the synthesis procedure, plenty of ZIF-8 nanoparticles (1.5 g) and only small amount of $\text{Ni}(\text{NO}_3)_2 \cdot 6\text{H}_2\text{O}$ (30 mg) were added into the precursor solution. What role do massive ZIF-8 nanoparticles play in the preparation of NiSA/PCFM membrane, pore former or other function?*

Reply and modification: Thank you for the question. ZIF-8 nanoparticles mainly acted as pore former in the preparation of NiSA/PCFM. Ni species were introduced by $\text{Ni}(\text{NO}_3)_2 \cdot 6\text{H}_2\text{O}$ and would be reduced to Ni single atoms by carbonized organic polymers of PAN in the following heated treatment.

Point 3: *Zn residues might still exist in NiSA/PCFM membrane, since ZIF-8 nanoparticles contain abundant Zn species. Moreover, Zn-based materials also show catalytic activity for CO₂ electroreduction (e.g., Angew. Chem. Int. Ed. 2016, 55, 9297–9300). Hence, contrast experiments should be performed.*

Reply and modification: This is a good question! According to your suggestion, we utilized multiple methods to detect the Zn species. Firstly, the energy-dispersive X-ray (EDX) spectroscopy elemental mappings in Figure 1e show the homogenous distributions of Ni, C, and N throughout the entire nanofiber architectures. On the contrary, the signal of Zn element is sparse and random in Figure R1a, which can not prove the existence of Zn element. In addition, Ni, C, and N peaks could be all observed in survey spectra of NiSA/PCFM in Figure R4b (or Figure S13), but there is no peaks indexed to Zn element, which should be centered around at 1021.6 and 1044.8 eV (Angew. Chem. 2019, 131, 3549-3553). Hence, there might be only trace (or no) Zn in NiSA/PCFM after being etched by H_2SO_4 , which might only have negligible impact on CO₂ electroreduction.

Figure R1. EDX mapping of Zn element (a) and survey XPS spectra (b) of NiSA/PCFM.

Point 4: Magnified HAADF-STEM images of NiSA/PCFM in Figure 1f displays the well-distributed Ni single atoms. Similar HAADF-STEM images of NiSA/CFM would be also welcomed for better understanding of the Ni species.

Reply and modification: Thank you for the question. According to your suggestion, we characterized NiSA/CFM catalysts via aberration-corrected HAADF-STEM. As shown in Figure R2, Ni element also shows atomic dispersion in the HAADF-STEM pattern of NiSA/CFM. We added this content in the supported information as Figure S8.

Figure R2. Aberration-corrected HAADF-STEM image of NiSA/CFM, those white dots are supposed to be Ni single atoms.

Point 5: Based on my understanding, Ni or NiO nanoparticles could also form under high-temperature carbonization because of the easy accumulation of Ni ions. Do Ni or NiO nanoparticles exist in NiSA/PCFM membrane?

Reply and modification: Thank you for the comment. We quite agree with your opinion that Ni or NiO nanoparticles might also emerged during high-temperature carbonization, due to the accumulation of Ni atoms. After heat treatment, the resultant materials were immersed into H₂SO₄ solution (3.0 M) for 10 hours to remove the remaining Ni or NiO nanoparticles. Furthermore, a variety of techniques were employed to characterize the as-synthesized NiSA/PCFM membrane. Firstly, no Co nanoparticles were observed in SEM or TEM images of NiSA/PCFM (Figure 1c-d,

Figure S4). The SAED (inset of Figure 1d) and XRD (Figure S9) patterns of NiSA/PCFM only proved the (002) and (100) planes of carbon. There are no characteristic peaks assigned to Ni or Ni oxides crystals in NiSA/PCFM. Moreover, the EDX mappings show the homogenous distributions of Ni over the carbon nanofiber. If there are plenty of highly dispersed Ni or NiO nanoparticles, they would be much easier seen than isolated Ni atoms in TEM images. Unfortunately, no obvious metallic clusters or nanoparticles were observed in any TEM images, although high-density and even-distributed Ni signals were displayed in EDX mapping. Hence, there might be only trace (or no) amount of Ni or Ni nanoparticles, which could not have significant impact on the catalytic activity of CO₂ electroreduction.

Point 6: *NiSA/PCFM shows lower electric resistance in EIS test. Why?*

Reply and modification: Thank you for the comment. Based on the orientation of the distorted graphene layers in HR-TEM images (Figure R3 or Figure S4d), a longer range orientation of the graphene unit is seen in NiSA/PCFM. These highly-graphitized graphene layers are beneficial for electron transfer, leading to lower electric resistance. On the contrary, only amorphous carbon was observed in the TEM images of NiSA/CFM (Figure S5), which is not in favor of the electron transfer through the catalyst surface.

Figure R3. A TEM image of NiSA/PCFM.

Point 7: *The description of the flow cell is very vague, and an enhanced discussion would be necessary, e.g., the flow rate of both electrolyte and CO₂ gas.*

Reply and modification: Thank you for the suggestion. The detailed description of the flow cell is presented in Figure R4. A commercial Pt/C electrode was used as anode and an Ag/AgCl were acted as the reference. 0.5 M KHCO₃ aqueous solution was utilized as electrolyte, which were separated by a piece of Nafion®117 membrane. Electrolyte were cycled at 50 mL min⁻¹. The CO₂ gas was supplied at rate of 20 mL min⁻¹ to the cathode and was flown through the NiSA/PCFM membrane.

Figure R4. The diagram of flow cell device used in our research.

Point 8: NiSA/PCFM powder was drop-casted onto a carbon paper using Nafion binders for H-type cell, and NiSA/PCFM membrane was directly used as a GDE device for CO₂ electroreduction. This comparison is not appropriate. Contrast experiment with NiSA/PCFM powder being drop-casted onto a GDE device would be necessary.

Reply and modification: Thank you for the suggestion. A specific volume of NiSA/PCFM ink was also drop-casted on a gas diffusion layer (SIGRACET) as the cathode compartment in flow cell for comparison, which was denoted as P-NiSA/PCFM. The CO FEs obtained with P-NiSA/PCFM was very close to those of NiSA/PCFM using cathode potentials from $-0.3 V_{\text{RHE}}$ to $-1.2 V_{\text{RHE}}$, reaching a maximum value of 87% at $-1.0 V_{\text{RHE}}$ (Figure R5a or Figure S21a). Furthermore, P-NiSA/PCFM could maintain a lower but steady current density at relatively higher cathode potential for 2 hours, e.g., 70.5 mA cm^{-2} at $-0.6 V_{\text{RHE}}$. However, current

densities dropped significantly at more negative $-0.8 V_{\text{RHE}}$ or $-1.0 V_{\text{RHE}}$ cathode potential within the short time of 1 hour (Figure R5b-d or Figure S21b-d), which is noticeably worse than the long-term stability of NiSA/PCFM. In a typical GDE device, the catalysts (e.g., P-NiSA/PCFM) were usually spray-coated (or deposited) onto a gas diffusion layer using polymer binders. During electrolysis procedure, massive gas products would pour out from the active sites along with the large current densities at relatively negative cathode potentials. Consequently, the electroactive species would easily fall off and reduce the long-term performance of CO_2 reduction, due to the weak contacts between catalyst and substrate.

Figure R5. Faradaic efficiencies of CO using NiSA/PCFM and P-NiSA/PCFM (a); stability tests of P-NiSA/PCFM in flow cell at $-0.6 V_{\text{RHE}}$ (b), $-0.8 V_{\text{RHE}}$ (c) and $-1.0 V_{\text{RHE}}$ (d).

Point 9: *The manuscript characterized and claimed excellent stability of NiSA/PCFM membrane after long-term testing, e.g., SEM, XPS, and TEM. Would NiSA/PCFM membrane fall apart after electrolysis?*

Reply and modification: Thank you for the comment. After 36 h of electrolysis, SEM, TEM, and XPS analysis of the NiSA/PCFM sample were conducted. As shown in Figure S22a-b (or Figure R6a-b), NiSA/PCFM still maintained its hierarchical porous structure there are no obvious metallic clusters or nanoparticles in SEM or TEM

images. The XPS spectrum of NiSA/PCFM still have the peak of N–Ni species, indicating the robust electrochemical stability and reusability of NiSA/PCFM.

Figure R6. Characterization of NiSA/PCFM after reuse: (a) SEM images, (b) TEM images; (c) Ni 2p and (d) N 1s XPS spectra.

Point 10: *Stability is one of the most important technique targets in CO₂ electroreduction. Both CO FEs and partial current densities of the NiSA/PCFM membrane showed only a negligible drop during 40 hours tests. It is an impressive but not fully-convincing result. Thus, stability test with longer time is welcomed.*

Reply and modification: Thanks for your suggestion. Stability tests with longer time were conducted using NiSA/PCFM catalysts at $-1.0 V_{RHE}$ electrode potential, and the reduction products were detected by GC for every 7 hour. CO FEs and partial current densities against time are plotted in Figure R7 or Figure 3e. Both CO FEs and partial current densities of the NiSA/PCFM membrane showed only a negligible drop during 120 hours tests, retaining more than 95% of initial value.

Figure R7. Long-term stability tests of NiSA/PCFM in GDE cell at $-1.0 \text{ V}_{\text{RHE}}$ cathode potential for 120 hours.

Thank you again for your warm and constructive comments improving the quality of this manuscript.

Response to the Reviewer 2

Point 1: Several experimental techniques have been reported in recent papers for the synthesis of single-atom Ni catalysts (like Chem 2018, 4, 285; Nat. Commun. 2019, 10, 2359). Author should provide the detailed formation mechanism of this NiSA/PCFM membrane.

Reply and modification: Thank you for the suggestion. The primary fibers were firstly electrospun from the mixture solution of ZIF-8 nanoparticles, $\text{Ni}(\text{NO}_3)_2 \cdot 6\text{H}_2\text{O}$ and polyacrylonitrile (PAN). Notably, ZIF-8 nanoparticles did not contain Ni ions and only acted as pore former in our experiments. Ni ions were introduced by $\text{Ni}(\text{NO}_3)_2 \cdot 6\text{H}_2\text{O}$ and would be reduced by carbonized organic polymers of PAN in the following heated treatment, leaving atomically dispersed Ni atoms anchored on nitrogen-doped porous carbon.

Point 2: Author mentioned that “NiSA/PCFM membranes yielded in our laboratory apparatus are at least $\sim 280 \text{ cm}^2$ in size”. Author should also offer the thickness and weight of this NiSA/PCFM membrane, which are equally important for an electro-catalyst

Reply and modification: Thank you for the comment. In our experiments, NiSA/PCFM membrane could be synthesized $\sim 280 \text{ cm}^2$ and $\sim 1 \text{ g}$ at one single time. NiSA/PCFM membrane is the multilayer structure with a total thickness of $\sim 0.5 \text{ mm}$.

Point 3: In a typical GDE device, electro-catalysts were mostly immobilized onto a conductive and porous substrate via polymer binders or electrodeposition, easily causing the fragility of the whole system. The integrated NiSA/PCFM membrane could be directly utilized as a GDE device, producing CO with more than -300 mA cm^{-2} partial current density. Author should also load NiSA/PCFM powder onto a gas-diffusion substrate, and compare the performance in CO_2 electroreduction.

Reply and modification: Thank you for the comment. A specific volume of NiSA/PCFM ink was also drop-casted on a gas diffusion layer (SIGRACET) as the

cathode compartment in flow cell for comparison, which was denoted as P-NiSA/PCFM. The CO FEs obtained with P-NiSA/PCFM was very close to those of NiSA/PCFM using cathode potentials from $-0.3 V_{\text{RHE}}$ to $-1.2 V_{\text{RHE}}$, reaching a maximum value of 87% at $-1.2 V_{\text{RHE}}$ (Figure R1a or Figure S21a). Furthermore, P-NiSA/PCFM could maintain a lower but steady current density at relatively higher cathode potential for 2 hours, e.g., 70.5 mA cm^{-2} at $-0.6 V_{\text{RHE}}$. However, current densities dropped significantly at more negative $-0.8 V_{\text{RHE}}$ or $-1.0 V_{\text{RHE}}$ cathode potential within the short time of 1 hour (Figure R1b-d or Figure S21b-d), which is noticeably worse than the long-term stability of NiSA/PCFM.

Figure R1. Faradaic efficiencies of CO using NiSA/PCFM and P-NiSA/PCFM (a); stability tests of P-NiSA/PCFM in flow cell at $-0.6 V_{\text{RHE}}$ (b), $-0.8 V_{\text{RHE}}$ (c) and $-1.0 V_{\text{RHE}}$ (d).

Point 4: NiSA/PCFM membrane could produce CO with a $-308.4 \text{ mA cm}^{-2}$ partial current density and 88% faradaic efficiency for up to 40 hours. This is indeed an excellent result. It is notable that other researchers have reported the constant ethylene selectivity for an initial 150 operating hours (Science 2018, 360, 783). Hence, stability test of NiSA/PCFM membrane for more than 100 hours should be conducted.

Reply and modification: Thank you for the comment. Stability tests with longer time were conducted using NiSA/PCFM catalysts at $-1.0 V_{\text{RHE}}$ electrode potential, and the reduction products were detected by GC for every 7 hour. CO FEs and partial current densities against time are plotted in Figure R2 or Figure 3e. Both CO FEs and partial current densities of the NiSA/PCFM membrane showed only a negligible drop during 120 hours tests, retaining more than 95% of initial value.

Figure R2. Long-term stability tests of NiSA/PCFM in GDE cell at $-1.0 V_{\text{RHE}}$ cathode potential for 120 hours.

Point 5: NiSA/PCFM membrane could be stably operated at industrial-relevant current densities. Would this catalysis system still be stable under large flow rate of CO_2 gas? Please clarify.

Reply and modification: Thank you for the comment. We have optimized various parameters of the flow cell, including the flow rate of CO_2 gas. Under optimized conditions, the CO_2 gas was supplied at rate of 20 mL min^{-1} to the cathode, producing CO with a 308.4 mA cm^{-2} partial current density and 88% faradaic efficiency for up to 120 hours. The slight change of CO_2 flow rate has negligible impact on the CO_2 reduction performance. However, the significant increase of CO_2 flow rate (e.g., $>30 \text{ mL min}^{-1}$) could cause the pressure imbalance between the gas/liquid phases around the active sites, leading to the poor stability of whole system.

Point 6: Flow cell reactor was introduced in this manuscript, which is also mentioned in recent works about CO₂ reduction (*Science* 2019, 365, 367; *Nat. Commun.* 2019, 10, 2807). Intuitive description of the flow cell reactor used in this manuscript should be provided.

Reply and modification: Thank you for the suggestion. Measurements at high current densities were performed in a self-made micro flow cell (Figure R3a or Figure S20a). Different from tests using H-type cell, NiSA/PCFM membranes were cut into the desired size and shape and directly served as cathode compartment, since these membranes are flexible and self-supporting. A commercial Pt/C electrode was used as anode and an Ag/AgCl were acted as the reference. 0.5 M KHCO₃ aqueous solution was utilized as electrolyte, which were separated by a piece of Nafion®117 membrane. In addition, electrolyte were cycled at with a rate of 50 mL min⁻¹. The CO₂ gas was supplied to the cathode and was flown through the NiSA/PCFM membrane at rate of 20 mL min⁻¹ (Figure R3b or Figure S20b).

Figure R3. Schematic diagram of GDE device (a) and the whole CO₂ electrolysis system (b).

Point 7: The pH value of electrolyte has significant impact on CO₂ electroreduction. Therefore, CO₂ electroreduction should also be tested in different electrolyte (like KCl, or KOH).

Reply and modification: Thank you for the suggestion. CO₂ electroreduction was tested in 0.5M KCl, 0.5M KHCO₃ and 0.5M KOH electrolyte using NiSA/PCFM catalyst as contrast. The CO and H₂ faradaic efficiencies are summarized in Figure R4. Under the same conditions, 0.5M KOH electrolyte could generate better overall CO faradaic efficiencies than 0.5M KCl, and 0.5M KOH. Especially, 86% and 81% CO faradaic efficiency were observed in 0.5M KCl (-1.1 V_{RHE}) and 0.5M KOH (-0.6 V_{RHE}) electrolyte.

Figure R4. CO and H₂ faradaic efficiencies in KCl (a), KHCO₃ (b) and KOH (c) electrolyte.

Thank you again for your warm and constructive comments improving the quality of this manuscript.

Response to the Reviewer 3

Thank you very much for reviewing our manuscript, and providing constructive questions, comments and suggestions! They are all very important to this paper. We really appreciate it!

***Point 1:** The fabrication novelty of Ni single atom is similar to previous reports. As we know, ZIF derived Ni single atom electrocatalyst for CO₂RR application has recently been published over three times [J. Am. Chem. Soc. 2017, 139, 8078-8081; J. Am. Chem. Soc. 2017, 139, 14889-14892; Energy Environ. Sci., 2018, 11, 1204-1210]. Besides, the self-supported single-atom Ni decorated carbon matrix or metal free carbon membrane were also published and shown an excellent CO₂RR performance [Joule, 2019, 3, 584-594, Angew. Chem. Int. Ed., 2017, 56, 7847-7852.]. Therefore, both of scientific concept and fabrication strategy are still undermined by the high similarity with previous reports.*

Reply and modification: Thank you for your comments. Just like you pointed out, several papers have reported Ni single atom as electrocatalyst for CO₂RR. In those reports, Ni ions were usually confined (or doped) within ZIFs and then reduced to single-atom Ni catalysts by the surrounding carbon ligands of ZIFs. However, massive Ni single atoms are embedded inside and can not work as effective active sites for CO₂RR.

In this manuscript, we firstly proposed a strategy to maximize the utilization of Ni single atoms in CO₂RR by constructing a free-standing and hierarchically porous carbon nanofiber membrane derived from polymers via electrospinning method. This unique structure could generate plenty of effective Ni single atoms, which actually participate in CO₂RR. Notably, ZIF-8 nanoparticles only acted as pore former in our experiments and Ni ions were introduced by Ni(NO₃)₂·6H₂O. In addition, the excellent mechanical strength and well-distributed Ni atoms of NiSA/PCFM successfully combine gas-diffusion and catalyst layers into one architecture for the first time. This integrated NiSA/PCFM membrane could be directly used as gas

diffusion electrode to establish an extremely stable three-phase interface for high-performance CO₂ electroreduction.

Furthermore, this fabrication strategy could also produce other single-atom metal catalyst, such as Co. CoSA/PCFM membrane was prepared directly from the mixture solution of Co(NO₃)₂·6H₂O, PAN, and ZIF-8 nanoparticles. CoSA/PCFM membrane could also be synthesized in relatively large scale, like NiSA/PCFM (Scheme 1b). Figure R1a represent a piece of ~300 cm² CoSA/PCFM with ultra-flexibility and mechanical stability, which is also highly desired for fabricating flexible working electrodes. As revealed in Figure R1b-d, CoSA/PCFM membrane single-atom Co structure was confirmed by XANES and EXAFS spectra.

Figure R1. Photos of a piece of flexible CoSA/PCFM membrane (a); XANES spectra at the Co K-edge of Co foil, Co₃O₄ and CoSA/PCFM (b-c); Fitting for EXAFS data of CoSA/PCFM, inset is the Co-N₄ structure (d).

Point 2: *The authors highlight the high current density of their electrocatalyst. I agree with the author's viewpoint that industrial-relevant current densities (>100 mA cm⁻²) is a significant point to judge the performance of CO₂RR electrocatalyst. However, this technologic problem has been solved by gas-diffusion cell system which has*

reported in many previous papers (at least 20 papers), such as, *Science*, 2018, 360, 783–787, *Energy Environ. Sci.*, 2019, 12, 1442-1453, *Adv. Mater.* 2018, 30, 1803111, *Energy Environ. Sci.*, 2019, 12, 1442-1453 et al.]. Therefore, this work has not contributed enough novelty in technological area of CO₂RR if it is just used in gas-diffusion cell system to achieve a high current density.

Reply and modification: This is a good question! Gas-diffusion cell system is one of the most promising techniques to obtain industrial-relevant current densities (>100 mA cm⁻²). To produce a usable gas-diffusion electrode, catalysts for CO₂ electroreduction were usually adhered or deposited on a gas-diffusion layer (Figure R2a, *Energy Environ. Sci.* 2019, 12, 1442-1453; *ACS Energy Lett.* 2019, 4, 639-643). The thickness of catalyst layer is always in nanoscale, e.g., 25 to 1000 nm in *Science*, 2018, 360, 783-787. This thin catalyst layer could easily be flooded by electrolyte and destroy the three-phase interface around the active sites, which is the key factor for high CO₂ reduction currents. Furthermore, because of the weak contacts between substrate and electro-catalyst, the electroactive species become detached and easily fall off especially under large currents, thus reducing the long-term performance of CO₂ electroreduction.

In our system, catalyst sites (Ni single atoms) were homogeneously distributed in gas-diffusion layer (porous carbon nanofibers) to get an integrated NiSA/PCFM membrane. The excellent mechanical strength, cross-linked carbon nanofibers and well-distributed Ni atoms of NiSA/PCFM successfully combine gas-diffusion and catalyst layers into one architecture (Figure R2b). This integrated NiSA/PCFM membrane could be directly used as gas diffusion electrode in flow cell device. The thickness of NiSA/PCFM membrane is in the range of hundreds of micrometers (Figure R2c), which is not easily to be totally soaked by electrolyte and establish an extremely stable three-phase interface for high-performance CO₂ electroreduction. This work may provide some guidance for the design integrated electro-catalysts utilized in gas-diffusion device.

Figure R2. Schematic for typical GDE cell with a catalyst loaded onto a gas-diffusion layer (a) and NiSA/PCFM membrane directly used as GDE (b); Cross-sectional SEM images of NiSA/PCFM (c).

Point 3: *It is noteworthy that the highest jco of NiSA/PCFM is 29.5 mA cm⁻² in H-type cell, which is quite similar to previous reports [J. Am. Chem. Soc. 2017, 139, 14889-14892, Adv. Mater. 2018, 30, 1706287]. Moreover, the comparison samples in table S4 were tested in H-type cell system but the jco of NiSA/PCFM in table S4 was obtained in gas-diffusion cell system. Therefore, it is unreasonable to claim that the NiSA/PCFM shows a higher CO₂RR performance because these values came from two different test system and mass loading.*

Reply and modification: Thank you for your suggestion! To evaluate and compare the catalytic activity of three samples in CO₂ reduction, NiSA/PCFM, NiSA/CFM and PCFM powders were firstly drop-casted onto a carbon paper using typical powdering and binding method for LSV tests and electrolysis in H-type cell. These powdered samples were renamed as P-NiSA/PCFM, P-NiSA/CFM and P-PCFM, easy for you to distinguish. The highest CO partial current density of P-NiSA/PCFM was 29.5 mA cm⁻². Notably, this integrated NiSA/PCFM membrane could be directly used as gas diffusion electrode in flow cell, and it also could directly acted as working electrode in

H-type cell system (Figure R3a). The free-standing and hierarchically porous carbon structure of NiSA/PCFM membrane could generate plenty of effective Ni single atoms, which actually participate in CO₂ reduction. As displayed in Figure R3b-c, the best CO Faradaic efficiency obtained by NiSA/PCFM is 96%, similar to that of P-NiSA/PCFM (95%). However, NiSA/PCFM brought forth a 56.1 mA cm⁻² CO partial current density at -1.0 V_{RHE}, nearly twice as much as that of P-NiSA/PCFM (29.5 mA cm⁻²). This content was added in the manuscript and Table S5.

Figure R3. Picture of a NiSA/PCFM cathode (a); CO Faradaic efficiencies (b) and CO partial current densities of two samples (c).

Besides, we have change the relevant content in manuscript into “To the best of our knowledge, accounting for the electrochemical conversion of CO₂ to CO, the catalytic activity of NiSA/PCFM is comparable to these state-of-the-art electro-catalysts in previously reports (Table S5), especially at industrial-relevant current densities”.

Point 4: *The authors should provide evidences to demonstrate the advantages of their NiSA/PCFM in stability and current density by additional control group. For instance, loading ZIF-derived NiSA porous carbon in commercial gas diffusion layer.*

Reply and modification: Thank you for your suggestion! NiSA/PCFM was powdered and drop-casted on a gas diffusion layer (SIGRACET) as the cathode compartment in flow cell for comparison, which was denoted as P-NiSA/PCFM. The CO FEs obtained with P-NiSA/PCFM was very close to those of NiSA/PCFM using cathode potentials from $-0.3 V_{\text{RHE}}$ to $-1.2 V_{\text{RHE}}$, reaching a maximum value of 87% at $-1.0 V_{\text{RHE}}$ (Figure R4a or Figure S21a). Furthermore, P-NiSA/PCFM could maintain a lower but steady current density at relatively higher cathode potential for 2 hours, e.g., 70.5 mA cm^{-2} at $-0.6 V_{\text{RHE}}$. However, current densities dropped significantly at more negative $-0.8 V_{\text{RHE}}$ or $-1.0 V_{\text{RHE}}$ cathode potential within the short time of 1 hour (Figure R4b-d or Figure S21b-d), which is noticeably worse than the long-term stability of NiSA/PCFM.

Figure R4. Faradaic efficiencies of CO using NiSA/PCFM and P-NiSA/PCFM (a); Stability tests of P-NiSA/PCFM in flow cell at $-0.6 V_{\text{RHE}}$ (b), $-0.8 V_{\text{RHE}}$ (c) and $-1.0 V_{\text{RHE}}$ (d).

According to your suggestion, we also prepared a Ni single atoms/porous carbon (NiSA/C) catalyst derived from Ni/ZIF-8 nanoparticles using the method in former reports (J. Am. Chem. Soc. 2017, 139, 8078–808). SEM images and EDS mapping of

NiSA/C indicate the ~500 nm size range with well distributed C, N and Ni element (Figure R5a-b). Aberration-corrected HAADF-STEM images and EXAFS spectra (Figure R5c-d) of NiSA/C demonstrated the existence of Ni single atoms.

Figure R5. The SEM (a), EDS mapping (b) and aberration-corrected HAADF-STEM (c) images of NiSA/C; EXAFS spectra at the Ni K-edge of Ni foil, NiO and NiSA/C (d); Faradaic efficiencies of CO using NiSA/PCFM and NiSA/C (e); Stability tests of NiSA/C in flow cell at $-1.0 V_{RHE}$ (f).

NiSA/C was drop-casted on a gas diffusion layer (SIGRACET) with same loading amount as NiSA/PCFM and worked as the cathode compartment in flow cell for CO₂ reduction. The CO FEs obtained with NiSA/C was lower than those of NiSA/PCFM

using cathode potentials from $-0.3 \text{ V}_{\text{RHE}}$ to $-1.2 \text{ V}_{\text{RHE}}$, reaching a maximum value of 83% at $-1.0 \text{ V}_{\text{RHE}}$ (Figure R5e). Moreover, current density at $-1.0 \text{ V}_{\text{RHE}}$ decreased obviously in less than two hours, revealing the poor stability of NiSA/C (Figure R5f).

Consequently, our NiSA/PCFM membrane combined gas-diffusion and catalyst layers into one architecture (see Figure R2b), which could be directly used as gas diffusion electrode, and has significant superiority to the typical catalyst/gas-diffusion structure (see Figure R2a) in the flow cell device.

Point 5: Generally, the calculation of CO₂RR faradic efficiency shows a $\pm 10\%$ errors, especially in gas-diffusion cell system. Therefore, it is necessary to add error bars for all faradic efficiency to ensure the accuracy of data.

Reply and modification: Thank you for your suggestion! Error bars have been added into all the faradic efficiencies.

Point 6: The LSV curves of NiSA/PCFM (NiSA/CFM, PCFM) tested in the gas-diffusion cell system should be provided (not just jco).

Reply and modification: Thank you for your suggestion! NiSA/PCFM, NiSA/CFM, and PCFM were powdered and drop-casted on a gas diffusion layer (SIGRACET) as the cathode compartment in flow cell for LSV tests. As presented in Figure S17d-e (or Figure R6), the current densities of three samples observed in flow cell are all far greater than those in H-type cell.

Figure R6. LSV tests of three samples in CO₂-saturated 0.5 M KHCO₃ solution in flow cell.

Point 7: As shown in supporting information, the authors use $Q=I \times t$ to calculate faradic efficiency. However, different with the sealing structure of H-type cell, for the gas diffusion cell, the gas products are fluxed to GC directly because the gas is flowing and products concentration does not relevant to the quantity of electricity. Therefore, please explain the measurement and calculate method of the faradic efficiency. The calculate formula of faradic efficiency in gas diffusion cell show as follows:

Reply and modification: Thank you for your suggestion! The electrolyzer outlet was vented directly into the gas-sampling loop of the gas chromatograph (GC, SRI-8610C, Mandel) equipped with a flame ionization detector (FID) and a thermal conductivity detector (TCD). The Faradaic efficiency (FE) of CO (or H₂) was calculated with the equation (Science, 2019, 365, 367-369; Science, 2018, 360, 783-787):

$$FE = n_{CO} * F * m_{CO} * V_m / I$$

Where n_{CO} is the number of electrons exchanged, $F=96,485$ C/mol, m_{CO} is the mole fraction of CO in the gaseous mixture analysed, V_m is the molar flow rate in mol/s, and I is the total current in A. The molar flow rate is derived from the volume flow rate V by the relation $V_m = pV/RT$. p is the atmospheric pressure in Pa, R is 8.314 J/mol K, and T is the temperature in K.

Point 8: The authors claim that the active center of NiSA/PCFM is Ni-N4 and they demonstrate it by EXAFS and HAADF-STEM image. But the HAADF-STEM image is not clear enough to identify Ni single atom. Please give a new image to ensure the Ni atom could be observed and measured. Does the scale of Ni atom in Figure 1f match its atomic diameter well?

Reply and modification: Thank you for your suggestion! The aberration-corrected HAADF-STEM was employed to directly observe those isolated Ni atoms on NiSA/PCFM catalyst. However, it is very difficult to make a reservation for aberration-corrected HAADF-STEM. We have done our best to get some new and clearer images (see Figure R7, or Figure 1f). the white dots have a diameter of ~ 0.261

nm, consistent with the radius of Ni atom (0.124 nm).

Figure R7. Aberration-corrected HAADF-STEM images of NiSA/PCFM, those white dots are supposed to be Ni single atoms.

Point 9: *The fabrication method of PCFM should be given. I have not found this part in both of manuscript and supporting information.*

Reply and modification: Thank you for your comments. The precursor solution containing 1.5 g of PAN and 1.5 g of ZIF-8 nanoparticles was also electrospun and carbonized under 900 °C, the resulting product was denoted to as PCFM. This content was added into the support information.

Point 10: *Many previous reports show that the heteroatom doped carbon materials (such as, nitrogen) also have outstanding CO₂RR performance in both of faradic efficiency and current density [Adv. Funct. Mater. 2018, 1800499, Adv. Energy Mater. 2017, 1701456, et al]. Therefore, the percentage of N with different configuration should be provided (based on XPS). Meanwhile, a control group should be given to exclude the main contribution of N if NiSA/PCFM (NiSA/CFM, PCFM) shows an obvious difference in N contents.*

Reply and modification: This is an excellent question. Carbon based single-atom metal catalysts (especially M-N-C structure) can not avoid the existence of N species. Just like you pointed out, N-doped carbon materials are also alternative catalysts for CO₂ electro-reduction. Therefore, multiple techniques were introduced to explore the dominant active sites in NiSA/PCFM. PCFM membrane does not own any single-atom Ni, but it has the similar nanostructure (Figure S8) and N content (Table

R1) as NiSA/PCFM. NiSA/PCFM could produce CO with 95% faradaic efficiency at $-0.7 V_{\text{RHE}}$. Under the same reaction conditions, the best CO faradaic efficiency achieved by PCFM is only 70% at a more negative cathode potential of $-0.8 V_{\text{RHE}}$ (Figure 3a). Moreover, the CO partial current densities of PCFM are also obviously lower than NiSA/PCFM (Figure 3b). Furthermore, DFT calculations (Figure 4a) demonstrate that the Ni-N₄-C structure has more favorable free energies (0.7 eV) than N-C structure (1.3 eV) for the generation of *COOH intermediate, consistent with the much smaller Tafel slope of NiSA/PCFM (117 mV/dec) than that of PCFM (204 mV/dec, Figure 4b). Therefore, single-atom Ni is the dominant active site for CO₂ reduction in our NiSA/PCFM catalyst, although N species is also in favor of this reaction.

Table R1 The element percentage of three samples.

	C (%)	N (%)	Zn (%)	Ni (%)
NiSA/PCFM	87.53	7.86	-	0.85
NiSA/CFM	89.03	7.82	-	0.92
PCFM	90.28	8.25	-	-

Point 11: *In the manuscript, the authors use 3 M H₂SO₄ solution to remove the Zn species. However, similar with Ni-N₄, the Zn-N_x may exist in the sample due to the high ratio of Zn in ZIF even though Zn is easy to volatilize in high temperature. Therefore, the amount of Zn should be measured by ICP-AES to exclude its contribution. The authors should provide additional DFT calculation with the Zn-N_x sites in CO₂RR if Zn content is too high to ignore.*

Reply and modification: Thank you for your comments. Only trace amount of Zn element was detected by ICP-AES in NiSA/PCFM catalyst, in accordance to XPS results (Table R1). Besides, EDX elemental mappings in Figure 1e show the homogenous distributions of Ni, C, and N throughout the entire nanofiber architectures. On the contrary, the signal of Zn element is sparse and random in Figure R8a, which can not prove the existence of Zn element. In addition, Ni, C, and N peaks could be all observed in survey spectra of NiSA/PCFM in Figure R8b, but there is no

peaks indexed to Zn element, which should be centered around at 1021.6 and 1044.8 eV (Angew. Chem. 2019, 131, 3549-3553). Hence, there might be only trace (or no) Zn in NiSA/PCFM after being etched by H₂SO₄, which might only have negligible impact on CO₂ electroreduction.

Figure R8. EDX mapping of Zn element (a) and survey XPS spectra (b) of NiSA/PCFM.

Point 12: It should be confirmed that the high CO₂RR performance of NiSA/PCFM comes from Ni single atom, as authors' declare, through additional metal poisoning experiments.

Reply and modification: Thank you for the comment. According to your suggestion, 50 mM KSCN was added into the CO₂-saturated 0.5 M KHCO₃ electrolyte. SCN⁻ ions could poison the Ni sites.

Figure R9. LSV tests of P-NiSA/PCFM in CO₂-saturated 0.5 M KHCO₃ or 0.5 M KHCO₃ + 50mM KSCN solution.

As displayed in Figure R9, the addition of 50 mM KSCN caused the sharp drop of current densities to about one tenth of the original value. Therefore, Ni single atoms are the dominant active sites for CO₂ electro-reduction.

Thank you again for your warm and constructive comments improving the quality of this manuscript.

Reviewers' comments:

Reviewer #1 (Remarks to the Author):

The authors have addressed all the comments from the three reviewers with substantial supplementary data. The manuscript has been improved and I believe it is suitable for publication in Nature Comm now.

Reviewer #2 (Remarks to the Author):

The authors have revised and improved the manuscript, but there are several major concerns and questions that need to be addressed and they are listed as follows. Only after addressing the following points, we can recommend its publication in Nat. Commun.

1. Stability test in flow-cell should be conducted in two-electrode system. The authors should provide more detailed information on the flow cell, such as the resistance, cell current at different applied cell voltage without IR correction (two-electrode system), the electricity-to-chemical efficiency of the total cell.
2. The description of the GDE is very vague, especially after long term stability tests. An enhanced characterization and discussion would be necessary.
3. In Flow cell electrolyzer, the gap between cathode and anode is impregnated with liquid electrolytes, which cause the electrolyzer has a large ohmic resistance. Due to the high ohmic loss at large current densities, the flow cell was limited to low efficiency of the energy. Thus, the authors should add more CO₂RR tests in Membrane Electrode Assembly (MEA) system. (Science, 2019, 364, 1091-1094; Science. 2019, 365, 367-369; Joule, 2019, 3, 265-278.)
4. Dilute streams necessitate energy-intensive downstream separations to purify the products and recover reactant. In this manuscript, the authors should conduct CO₂RR tests at different CO₂ gas flow rates. In the meanwhile, additional data (the concentration of the product streams and single pass conversion efficiency at different applied potentials and electrolyzers) should be given.
5. The authors claimed that "Certain amount of Nafion solution (5 wt%, Dupond) was spray-coated on this membrane to get a hydrophobic property". However, the hydrophobic polymer binder led to a significant decrease in electrochemically active surface area (ECSA) on the introduction of hydrophobicity. (Nature Materials. 2019, 18, 1222-1227) The electrochemically active surface areas of wettable and hydrophobic electrode should be provided, respectively.
6. The percentage of N with different configuration should be provided (based on XPS). The reviewer pointed out the configuration of N (The individual atomic concentrations of pyridinic, pyrrolic, and graphitic N atoms), but the authors seem skipped this point in your response letter.
7. In Table S5, the latest published articles should also be included as a comparison. (Science, 2019, 364, 1091-1094; Science. 2019, 365, 367-369; Energy Environ. Sci., 2019, 12, 2455; Joule, 2019, 3, 584-594; Joule, 2019, 3, 265-278. et al.) Besides, some important parameters, such as potential, medium and cathode material should be given to provide more comprehensive comparison to these reported electrocatalysts.

Reviewer #3 (Remarks to the Author):

The authors have fully addressed all of my concern with satisfied supplementary experiments and correct revisions. I am happy to recommend its publication in Nat Comm in its current version.

Response to Reviewer's Comments

Response to the Reviewer 2

Special thanks to you for your kind and professional comments, questions and suggestions! They are all very important to improve this manuscript and our future work. We really appreciate it!

Point 1: *Stability test in flow-cell should be conducted in two-electrode system. The authors should provide more detailed information on the flow cell, such as the resistance, cell current at different applied cell voltage without IR correction (two-electrode system), the electricity-to-chemical efficiency of the total cell.*

Reply and modification: Thank you for your comments. To measure the resistance of flow cell, we conducted an electrochemical impedance spectroscopy (EIS) test using an NiSA/PCFM membrane. As shown in Figure R1, the cell resistance cell was measured to be 1.25 Ω .

Figure R1. Impedance spectra of NiSA/PCFM membrane in flow cell.

According to your suggestions, we also conducted CO₂ electrolysis and stability

test of NiSA/PCFM membrane in a flow cell device using two-electrode system without IR compensation. The range of constant current density was from 100 to 300 mA cm⁻². As revealed in Figure R2a, the CO faradic efficiencies reached a maximum value of 91% at 250 mA cm⁻² current density. 90% CO faradic efficiency could still be obtained at 300 mA cm⁻² current density. The full-cell voltage increased from 3.25 V to 3.88 V along with the improvement of current density (Figure R2b).

In addition, the Energy efficiency was calculated with the formula (Science, 2019, 365, 367; Nature, 2019, 10.1038/s41586-019-1782-2):

$$\text{Energy efficiency} = (E^{\circ} \times \text{FE}_{\text{CO}}) / E_{\text{cell}}$$

FE_{CO} is the Faradaic efficiency of CO, E_{cell} is the full-cell voltage. E^o was calculated with the formula: E^o = E_{O₂} - E_{CO} = 1.23 V - (-0.11 V) = 1.34 V, where E_{CO} is the CO₂/CO equilibrium potential (Nature, 2016, 537, 382), E_{O₂} is the equilibrium potential of oxygen evolution reaction (Nat. Energy 2019, 4, 329). The calculated energy efficiencies for CO were displayed in Figure R2c, and a maximal 33.8% energy efficiency was observed at 200 mA cm⁻² current density.

Figure R2. CO₂ electrolysis of NiSA/PCFM membrane in two-electrode system. (a) CO faradic efficiency, (b) full-cell voltage, and (c) energy efficiency with different current densities; (d) Stability test at 250 mA cm⁻² current density.

Long-term electrolysis was performed using NiSA/PCFM membrane at a constant 250 mA cm^{-2} current density to investigate the stability. The CO faradic efficiency and cell voltage were detected during the test. As displayed in Figure R2d, both CO faradic efficiency and cell voltage of NiSA/PCFM membrane only declined slightly after 100 hours test, maintained more than 90% of initial value. This content was added into supported information as Figure S21.

Point 2: *The description of the GDE is very vague, especially after long term stability tests. An enhanced characterization and discussion would be necessary.*

Reply and modification: Thank you for your suggestions! As shown in Figure R3a, we constructed a more detailed graphic illustration of our GDE device. It is notable that NiSA/PCFM successfully combine gas-diffusion and catalyst layers into one architecture. This integrated NiSA/PCFM membrane could be directly used as a gas diffusion electrode without powdering and loading onto an extra gas-diffusion layer (Figure R3b-c). This content was added into manuscript as Figure 4c-e.

Figure R3. (a) Graphic illustration of a GDE device; Schematic for (b) NiSA/PCFM membrane directly used as GDE and (c) typical GDE cell with catalyst powder loaded onto a gas-diffusion layer via polymer binder.

After long-time stability test, multiple methods were conducted to explore the structure and composition of NiSA/PCFM, including SEM, TEM, XPS, XRD, and XANES analysis. As revealed by the SEM and TEM images in Figure R4, NiSA/PCFM retained network-like structure and well-distributed hollow nanocages within the carbonaceous nanofibers.

Figure R4. (a-b) SEM and (c-d) TEM images of NiSA/PCFM after electrolysis.

According to XPS spectra of NiSA/PCFM (Figure R5a-b), the Ni $2p_{3/2}$ peak still locates between metallic Ni⁰ (853.5 eV) and Ni²⁺ (855.8 eV), suggesting that the Ni atoms in NiSA/PCFM is likely to be in a low-valent state. XRD pattern of NiSA/PCFM (Figure R5c) shows only two peaks centered at 26.2° and 44.0°, indexed to the (002) and (100) planes of carbon. No diffraction peaks of metallic Ni or Ni oxides are observed. Moreover, Figure R5d exhibits the XANES spectrum in the Ni K-edge of NiSA/PCFM, using Ni foil and NiO as the references. The near-edge spectra of NiSA/PCFM after electrolysis locates between Ni foil and NiO, implying that the valence state of those isolated Ni atoms are between metallic (Ni⁰) and oxidized (Ni²⁺) status. Therefore, we can conclude that the Ni single atoms did not accumulate into nanoparticles during electrolysis. This content was added into supported information as Figure S23-24.

Figure R5. (a-b) XPS, (c) XRD and (d) XANES spectra of NiSA/PCFM after electrolysis.

Point 3: *In Flow cell electrolyzer, the gap between cathode and anode is impregnated with liquid electrolytes, which cause the electrolyzer has a large ohmic resistance. Due to the high ohmic loss at large current densities, the flow cell was limited to low efficiency of the energy. Thus, the authors should add more CO₂RR tests in Membrane Electrode Assembly (MEA) system. (Science, 2019, 364, 1091-1094; Science. 2019, 365, 367-369; Joule, 2019, 3, 265-278.)*

Reply and modification: Thank you for your suggestion! We have carefully studied the three papers, which are very illuminating and cited all of them in our manuscript. As presented in Figure R6a, reference1 (Science, 2019, 364, 1091-1094) used a similar flow cell device as our research, while zero-gap membrane flow reactors were utilized in reference2 (Science. 2019, 365, 367-369) and reference3 (Joule, 2019, 3, 265-278). In our former works, we did assemble a similar zero-gap membrane flow reactor as reference2 and reference3 (Figure R6d). According to your suggestions, we estimated the catalytic activity of NiSA/PCFM in CO₂ reduction using this MEA device. A NiSA/PCFM membrane was also directly used as cathode and RuO₂ was

used as anode material, which were separated by an anion exchange membrane. The CO₂ gas was humidified before venting into cathode compartment. 0.5 M KHCO₃ aqueous solution was flowed and recycled through the anode compartment.

[Redacted]

Figure R6. (a-c) Graphic illustration of the electrolyzers used in reference 1-3; (d) A picture of the membrane flow reactor in our work.

The range of constant current density was from 100 to 300 mA cm⁻². As revealed in Figure R7a, the CO faradic efficiencies obtained a maximum value of 84% at 150 mA cm⁻² current density. The full-cell voltage rose from 2.65 V to 3.08 V along with the increase of current density from 100 to 300 mA cm⁻² (Figure R7b). Notably, the full-cell voltage observed in the membrane flow reactor is significantly lower than that of flow cell at the same current density, e.g., 2.82 V for membrane flow reactor and 3.48 V for flow cell at 200 mA cm⁻² current density. This enhanced activity could be attributed to the zero-gap structure of membrane flow reactor, which cause the electrolyzer has a smaller ohmic resistance. Unfortunately, the faradic efficiencies in membrane flow reactor were lower than those in flow cell throughout the whole

current density range. We would furtherly study the membrane flow reactors and optimize the reaction conditions to realize high-performance CO₂ electro-reduction.

Figure R7. CO₂ electrolysis of NiSA/PCFM membrane in a membrane flow reactor using two-electrode system. (a) CO faradic efficiency and (b) full-cell voltage at different current densities.

Point 4: Dilute streams necessitate energy-intensive downstream separations to purify the products and recover reactant. In this manuscript, the authors should conduct CO₂RR tests at different CO₂ gas flow rates. In the meanwhile, additional data (the concentration of the product streams and single pass conversion efficiency at different applied potentials and electrolyzers) should be given.

Reply and modification: Thank you for your comments! We conducted CO₂ electrolysis using different CO₂ gas flow rates. A NiSA/PCFM membrane was used as cathode compartment with a constant current density of 100 or 200 mA cm⁻². Single-pass conversion efficiency was calculated via the CO production rate divided by CO₂ gas flow rate (mL min⁻¹). As revealed in Figure R8, CO₂ gas flow rate did not have significant impact on the CO faradic efficiencies at a constant current density of 100 mA cm⁻². CO faradic efficiencies increased slightly with the improvement of CO₂ gas flow rate at 200 mA cm⁻² current density. On the contrary, the single-pass conversion efficiency decreased sharply, when the CO₂ gas flow rate increased from 10 to 100 mL min⁻¹ at 100 and 200 mA cm⁻². Just as you pointed out, high CO₂ gas flow rates might cause the low concentration of the products and purification problems. For example, the volume ratio of CO/CO₂ is only 2.2/97.8 at 100 mA cm⁻²

with a mL min^{-1} CO_2 gas flow rate 100. Besides, high CO_2 gas flow rates might also result in the fragile of the cathode compartment and influence the long-time stability.

Figure R8. Faradaic efficiencies of CO and single pass conversion using different CO_2 flow rates at constant current density of (a) 100 and (b) 200 mA cm^{-2} .

Point 5: The authors claimed that “Certain amount of Nafion solution (5 wt%, Dupond) was spray-coated on this membrane to get a hydrophobic property”. However, the hydrophobic polymer binder led to a significant decrease in electrochemically active surface area (ECSA) on the introduction of hydrophobicity. (Nature Materials. 2019, 18, 1222–1227) The electrochemically active surface areas of wettable and hydrophobic electrode should be provided, respectively.

Reply and modification: Thank you for your suggestion! The ECSAs of wettable NiSA/PCFM and NiSA/PCFM coating with Nafion were determined by double-layer capacitance. According to Figure R9, the wettable NiSA/PCFM has larger ECSA (20.2 mF cm^{-2}) than that of NiSA/PCFM coating with Nafion (13.2 mF cm^{-2}), consistent with the former report (Nature Materials, 2019, 18, 1222-1227). However, the hydrophobicity is a crucial factor for GDE device. Once the gas diffusion layer floods, the pathways for the diffusion of CO_2 in the gas phase toward the catalyst become obstructed and high CO_2 reduction currents can no longer be sustained (Science 2018, 360, 783-787). In our research, certain amount of Nafion solution was spray-coated on NiSA/PCFM membrane to get a hydrophobic property and prevented NiSA/PCFM membrane from the easily flooding of electrolyte, which could provide a stable three-phase interface for long-term CO_2 electrolysis.

Figure R9. Double-layer capacitance of (a-b) wettable NiSA/PCFM and (c-d) NiSA/PCFM coating with Nafion.

Point 6: The percentage of N with different configuration should be provided (based on XPS). The reviewer pointed out the configuration of N (The individual atomic concentrations of pyridinic, pyrrolic, and graphitic N atoms), but the authors seem skipped this point in your response letter.

Reply and modification: Thank you for your comments! The atomic concentrations of pyridinic, pyrrolic, and graphitic N atoms were displayed in Table R1.

Table R1 The percentage of different configuration of N.

	Pyridinic N (%)	Pyrrolic N (%)	Graphitic N (%)
NiSA/PCFM	1.74	1.68	1.32
NiSA/CFM	1.92	1.82	1.08
PCFM	2.63	2.81	1.62

Point 7: In Table S5, the latest published articles should also be included as a comparison. (*Science*, 2019, 364, 1091-1094; *Science*, 2019, 365, 367-369; *Energy Environ. Sci.*, 2019, 12, 2455; *Joule*, 2019, 3, 584-594; *Joule*, 2019, 3, 265-278. *et al.*) Besides, some important parameters, such as potential, medium and cathode material should be given to provide more comprehensive comparison to these reported electrocatalysts.

Reply and modification: Thank you for your suggestion! These latest published articles were added into Table S5. Potential, medium and cathode material were given in Table S5 (or Table R2) for comprehensive comparison to these reported electrocatalysts.

Table S5. Electrocatalytic performance of recently reported single-atom catalysts for CO₂RR.

Cathode Materials	Atom%	Potential (vs.RHE)/V	Current Density/ mA cm ⁻²	Faradaic Efficiency /%	Products	Working Media	Loading/ mg cm ⁻²
This work NiSA/PCFM	1.3	-1.0 (GDE) or -0.7 (H-type)	308.4 (GDE) or 56.1 (H-type)	88 (GDE) or 96 (H-type)	CO	0.5 M KHCO₃	1.0
Ni SAs/N-C ^{S7}	1.53	-0.89	10.48	71.9	CO	0.5 M KHCO ₃	0.2
Ni-N ₄ -C ^{S8}	1.41	-0.81	28.6	99.0	CO	0.5 M KHCO ₃	0.02
Ni-NG ^{S9}	0.44	-0.62	11	95.0	CO	0.5 M KHCO ₃	1.0
NiN-GS ^{S10}	-	-0.82	20	93.2	CO	0.5 M KHCO ₃	1.0
A-Ni-NSG ^{S11}	0.95	-0.61	22	98.0	CO	0.5 M KHCO ₃	0.1
NiSA-N-CNTs ^{S12}	20	-0.70	23.5	91.3	CO	0.5 M KHCO ₃	0.2
Ni-SnS ₂ ^{S13}	5	-0.90	19.6	93.0	formate/ CO	0.1 M KHCO ₃	0.5
SE-Ni SAs@PNC ^{S14}	-	-1.00	18.3	88	CO	0.5 M KHCO ₃	0.4
Ni SAs/NCNTs ^{S15}	6.63	-0.75	20	95	CO	0.5 M KHCO ₃	0.8

Co-N ₂ ^{S16}	-	-0.52	18.1	94.0	CO	0.5 M KHCO ₃	0.8
Co-N ₅ ^{S17}	3.54	-0.79	10.2	99.0	CO	0.2 M NaHCO ₃	-
M-N ₄ ^{S18}	-	-0.29/-0.38	-	93.0/45.0	CO	0.1 M KHCO ₃	0.6
C-AFC@ZIF-8 ^{S19}	0.72	-0.63	10	93.0	CO	1 M KHCO ₃	2.0
ZnN _x /C ^{S20}	0.1	-0.43	4.8	95.0	CO	0.5 M KHCO ₃	0.4
Sn-CF1000 AD-Sn/N-C1000 S21	1.0	-0.8	11/5.12	62/91	formate/ CO	0.5 M KHCO ₃ /0.1 M KHCO ₃	0.97
Ni ₃ N/C ^{S22}	-	-0.85	12	92.5	CO	0.1 M KHCO ₃	0.75
Ni/N-CHSS ^{S23}	-	-0.9	15	93.1	CO	0.5 M KHCO ₃	0.5
Ni-NC-ATPA@ C ^{S24}	-	-0.7	6	93	CO	0.5 M KHCO ₃	-
Fe ³⁺ -N-C/GDE S25	-	-0.45	94	>90	CO	0.5 M KHCO ₃	2.5
CoPc ^{S26}	-	-	150	>95	CO	1 M KOH	-
CoPc ^{S27}	-	-	200	>90	CO	water	0.2
Ni-NCB ^{S28}	0.27	-	100	100	CO	0.5 M KHCO ₃	1.25
H-CPs ^{S29}	-	-1.0	48.66	97	CO	0.5 M KHCO ₃	-

Thank you again for your warm and constructive comments improving the quality of this manuscript.

REVIEWERS' COMMENTS:

Reviewer #2 (Remarks to the Author):

The authors have fully addressed all of comments with satisfied supplementary experiments and correct revisions. I am happy to recommend its publication in Nature Comm.